# Merging LoRAs like Playing LEGO: Pushing the Modularity of LoRA to Extremes Through Rank-Wise Clustering

**Ziyu Zhao[12], Tao Shen[1], Didi Zhu[1], Zexi Li[1], Jing Su[3], Xuwu Wang[3], Fei Wu[1]***
[1]Zhejiang University, [2]Shanghai Innovation Institute, [3]ByteDance Inc.
`benzhao.styx@gmail.com  wufei@zju.edu.cn`,

## Abstract

Low-Rank Adaptation (LoRA) has emerged as a popular technique for fine-tuning large language models (LLMs) to various domains due to its modular design and widespread availability on platforms like Huggingface. This modularity has sparked interest in combining multiple LoRAs to enhance LLM capabilities. However, existing methods for LoRA composition primarily focus on task-specific adaptations that require additional training, and current model merging techniques often fail to fully leverage LoRA's modular nature, leading to parameter interference and performance degradation. In this paper, we investigate the feasibility of disassembling and reassembling multiple LoRAs at a finer granularity, analogous to assembling LEGO blocks. We introduce the concept of Minimal Semantic Units (MSUs), where the parameters corresponding to each rank in LoRA function as independent units. These MSUs demonstrate permutation invariance and concatenation-summation equivalence properties, enabling flexible combinations to create new LoRAs. Building on these insights, we propose the LoRA-LEGO framework. This framework conducts rank-wise parameter clustering by grouping MSUs from different LoRAs into $k$ clusters. The centroid of each cluster serves as a representative MSU, enabling the assembly of a merged LoRA with an adjusted rank of $k$. Additionally, we apply a dual reweighting strategy to optimize the scale of the merged LoRA. Experiments across various benchmarks demonstrate that our method outperforms existing approaches in LoRA merging.

## 1 Introduction

Large Language Models (LLMs) like ChatGPT Achiam et al. (2023) and LLaMA Touvron et al. (2023) trained on vast amounts of general data, demonstrate remarkable performance in general tasks. To explore their potential for specialized tasks, adapting LLMs to specific domains by fine-tuning model parameters has become a critical area of research. In this context, Low-rank Adaptation (LoRA) Hu et al. (2021), as a parameter-efficient fine-tuning approach, has gained widespread recognition, also attributed to its modular design Liu et al. (2023); Yang et al. (2023b); Hadi et al. (2023). The modular nature of LoRA enables it to serve as ***plug-and-play*** plugins for LLMs, facilitating the storage and deployment of large collections of LoRAs on platforms like Hugging Face. The extensive availability of LoRAs has sparked considerable interest in combining multiple LoRAs into a unified adapter to significantly extend the capabilities of LLMs Yadav et al. (2024a); Xiao et al. (2024); Chen et al. (2025); Lyu et al. (2024); Zhu et al.; Huang et al. (2023).

Previous methods for composing multiple LoRAs have primarily focused on assembling separate LoRAs tailored to specific downstream tasks, which generally require additional training Wu et al. (2023); Wang et al. (2024); Chronopoulou et al. (2023); Yadav et al. (2024a); Huang et al. (2023). Model merging Tang et al. (2024); Yadav et al. (2024b); Ilharco et al. (2022); Yang et al. (2024a;b; 2025) offers an alternative approach by aggregating the parameters of multiple LoRAs into a unified adapter without extra training, producing a unified LoRA with comprehensive capabilities. However, these methods typically employ element-wise parameter fusion, which can neglect and disrupt the

---

*  Corresponding authors.

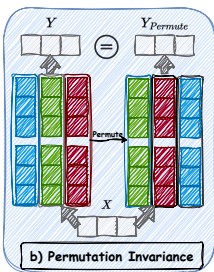 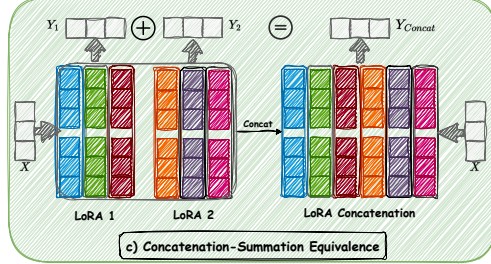

Figure 1: **Further Modularization of LoRA**: a) Each LoRA can be further modularized into multiple Minimal Semantic Units (MSUs), each corresponding to a row in $A$ matrix and a column in matrix $B$, differentiated by distinct colors. b) The MSUs within a LoRA display permutation invariance, implying that any rearrangement of the MSUs does not affect the output generated by the LoRA. c) Multiple LoRAs exhibit Concatenation-Summation Equivalence, indicating that the summation of outputs from various LoRAs is equivalent to the output of a singular LoRA constructed by concatenating their MSUs.

internal semantic structure within LoRA. This disruption potentially leads to parameter interference (as discussed in §2.3), thereby hindering the performance of merged LoRA. *This paper approaches LoRA merging from a novel perspective, focusing on the fine-grained modularization of LoRA by decomposing it into independent units, which enables the flexible reconstruction of a unified LoRA with comprehensive capabilities.*

As illustrated in Fig.1, our motivation for further modularizing LoRA stems from the following insights: a) Each rank in LoRA corresponds to a row in the down-projection matrix $A$ and a column in the up-projection matrix $B$ Zhao et al. (2025). Since the calculations for each rank are independent, we consider the parameters associated with each rank as a cohesive entity. We define these entities as **Minimal Semantic Units (MSUs)**, which serve as the fundamental building blocks of LoRA. b) Within each LoRA, the MSUs exhibit the property of **Permutation Invariance**, indicating that any permutation of MSUs within a LoRA does not affect the adapter's output. c) LoRA exhibits the **Concatenation-Summation Equivalence** property, which states that summing the outputs from multiple LoRAs is equivalent to the output of a single higher-ranked LoRA constructed by concatenating all the MSUs of these LoRAs.

In this paper, we introduce a novel method called LoRA-LEGO, which is based on the insight that MSUs act as building blocks that form a LoRA and can be disassembled and reassembled like playing with LEGO. LoRA-LEGO consists of three main steps: (1) **Grouping** MSUs from candidate LoRAs into a MSU pool; (2) **Clustering** the MSU pool into $k$ clusters, where $k$ is the target rank of the merged LoRA; (3) **Constructing** the merged LoRA from the centroids of these clusters, with each centroid representing an MSU, thereby setting the merged LoRA's rank to $k$. LoRA-LEGO enables the flexible combination of LoRAs with arbitrary ranks by clustering similar MSUs, at the same time effectively resolving parameter interference while merging. This approach allows for targeted rank adjustments in the merged LoRA to preserve task-specific knowledge. We also observed that variations in parameter norms and the rank size of the merged LoRA affect the output scale. To address this, we implement a dual reweighting strategy that adjusts both the parameters and the outputs, ensuring optimal scaling for the merged LoRA.

We empirically validate the effectiveness of the proposed LoRA-LEGO in both multi-task Tang et al. (2024) and mixed-task Zhao et al. (2024b) scenarios. Experimental results show that LoRA-LEGO consistently outperforms other methods for LoRA merging, demonstrating notable flexibility and efficiency. Additionally, LoRA-LEGO can merge heterogeneous LoRAs of varying ranks, surpassing the capabilities of previous model merging methods. Moreover, it can also be applied to individual LoRAs for parameter pruning, revealing that retaining just 50% of the parameters can achieve performance comparable to the original model. Our contribution can be summarized as:

- We investigate the modularization of LoRA, identifying the MSU as its fundamental building block, which is characterized by permutation invariance and concatenation-summation equivalence properties.

- We introduce LoRA-LEGO that merges multiple LoRAs in a LEGO-like fashion by grouping, clustering, and reconstructing MSUs to seamlessly combine separate LoRAs.

- Experimental results show that LoRA-LEGO can flexibly disassemble and reassemble LoRAs of any rank, surpassing other model merging methods in performance. Additionally, LoRA-LEGO can be effectively applied to individual LoRAs, enabling parameter pruning and a substantial reduction in LoRA parameters while maintaining comparable performance.

## 2 PRELIMINARIES

### 2.1 LOW-RANK ADAPTATION

Directly fine-tuning LLMs with full parameters is computationally intensive and is not feasible in low-resource scenarios. Based on the idea that only a small number of low-rank parameters need to be fine-tuned for sufficient performance in new domains, Hu et al. (2021) proposed the Low-Rank Adaptation, where the LoRA module can be combined with the pre-trained parameters in parallel for efficient inference.

Specifically, given pre-trained weights $W_0 \in \mathbb{R}^{d \times k}$ of a sub-module of LLM, the LoRA adds an extra trainable weight matrix as $W_0 + \Delta W = W_0 + BA$, where $\Delta W$ can be decomposed into two smaller matrices $B \in \mathbb{R}^{d \times r}$ and $A \in \mathbb{R}^{r \times d}$, where $r$ stands for the rank of $\Delta W$ and the rank $r \ll d$. The forward pass for a layer $y = W_0 x$ can be modified as follows:

$$y = W_0 x + \Delta W x = W_0 x + BA x, \tag{1}$$

where $x \in \mathbb{R}^d$ is the input and the $y \in \mathbb{R}^d$ denote the output.

### 2.2 FURTHER MODULARIZATION OF LORA

Before delving into the issue of LoRA merging, it is imperative to present several pivotal insights and definitions that could serve as fundamental components for constructing a LoRA module.

**Definition 1.** *Minimum Semantic Unit of LoRA. Let $A$ and $B$ be matrices in a LoRA module. For each index $i$, define the **minimum semantic unit** of LoRA as the combined vector $s_i = [a_i, b_i]$, where $a_i$ is the $i$-th row of $A$ and $b_i$ is the $i$-th row of $B^T$ (i.e., the transpose of the $i$-th column of $B$).*

In this context, each row of the down-projection matrix $A$ and its corresponding column in the up-projection matrix $B$ are treated as a cohesive unit, defined as a Minimum Semantic Unit (MSU). Each MSU contributes to a rank of the LoRA, encapsulating a distinct semantic fragment of the LoRA's capacity. Through this definition, LoRAs exhibit the following properties.

**Property 2.1.** *Permutation Invariance. For a LoRA module parameterized by matrices $A$ and $B$, if the rows of $A$ are permuted, then by performing a corresponding permutation of the columns of $B$, the product of these matrices remains unchanged. Formally, let $P$ be a permutation matrix that satisfies $P^T P = I$, where $I$ is the identity matrix. If we permute the rows of $A$ to obtain a new matrix $A' = PA$, and correspondingly permute the columns of $B$ to get $B' = BP^T$, then, $BA = B'A'$.*

The property of permutation invariance indicates that the arrangement of MSUs within LoRA calculations can be altered without affecting LoRA's output.

**Property 2.2.** *Concatenation-Summation Equivalence. Consider two LoRAs, $(A_1, B_1)$ and $(A_2, B_2)$, each of rank $r$. Specifically, matrices $A_1$ and $A_2$ are of size $\mathbb{R}^{r \times d}$, and $B_1$ and $B_2$ are of size $\mathbb{R}^{d \times r}$. Define the concatenated matrices as:*

$$A' = \begin{bmatrix} A_1 \\ A_2 \end{bmatrix} \in \mathbb{R}^{2r \times d}, \quad B' = [B_1 \quad B_2] \in \mathbb{R}^{d \times 2r}.$$

*The output vector $y$ from the concatenated model is equivalent to the sum of the outputs from each individual LoRA model:*

$$y = B'A'x = (B_1 A_1 + B_2 A_2)x.$$

Based on this property, we can synthesize the knowledge from all LoRAs by constructing a new LoRA through the concatenation of all MSUs from each LoRA. The computational result is equivalent to ensembling the outputs of all LoRAs. Based on these insights, we can draw the following conclusions:

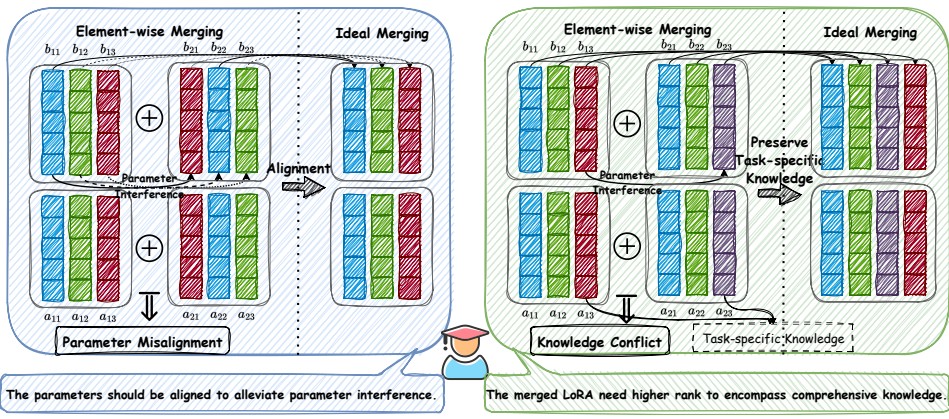

Figure 2: Two sources of parameter interference in LoRA merging. The left part illustrates how parameter misalignment can lead to interference; the right part demonstrates that knowledge conflict in merged LoRA layers can also result in parameter interference.

> Each LoRA can be modularized into multiple MSUs, with each MSU corresponding to a rank within the LoRA. These MSUs can be flexibly permuted and combined to construct a unified LoRA.

## 2.3 PROBLEM FORMULATION AND CHALLENGES

Consider a LLM denoted as $\mathcal{L}$ and a set of $p$ task-specific LoRAs, represented by $\Phi = \{\phi_1, \phi_2, \ldots, \phi_p\}$. Each LoRA $\phi_i$ is specialized for a particular task $\mathcal{T}_i$ and is crafted by incorporating low-rank matrices into different layers of $\mathcal{L}$, thereby tuning the model to better suit $\mathcal{T}_i$. For simplicity of notation, we denote the parameters of these low-rank matrices at any given layer for each LoRA $\phi_i$ as $A_i$ and $B_i$. The goal of merging these LoRAs is to synthesize a comprehensive LoRA $\phi'$ that not only excels in all tasks encompassed by $\Phi$ but also generalizes well to unseen tasks. We discuss the difference between the LoRA merging setting and the previous model merging setting in the Appendix A.

A natural approach to performing LoRA merging involves a simple element-wise averaging of the parameters from each LoRA: $\phi' = \frac{1}{p} \sum_{i=1}^{p} \phi_i$. However, **parameter interference** poses a significant challenge to effective LoRA merging. We identify two potential sources of parameter interference during LoRA merging and demonstrate through experiments that such interference can lead to performance degradation in the merged LoRA.

Table 1: **Performance degradation after merging misaligned LoRAs.** "Original" refers to the performance of the unaltered LoRA, while "Misaligned" indicates the performance after merging the LoRA with a randomly permuted version of itself.

| Task | Original | Misaligned |
|---|---|---|
| CoLA | **61.63** | 60.96 (1.1% ↓) |
| MNLI | **77.46** | 69.49 (10.3% ↓) |
| MRPC | 68.00 | **68.50** (-0.7% ↓) |
| QNLI | **77.25** | 60.44 (21.8% ↓) |
| QQP | **75.83** | 66.94 (11.7% ↓) |
| RTE | 52.22 | **54.44** (-4.2% ↓) |
| SST2 | **75.74** | 75.52 (0.3% ↓) |
| Overall | **69.73** | 65.18 (5.74% ↓) |

The first cause of parameter interference stems from **parameter misalignment** in LoRAs, as depicted in the left part of Fig.2. According to Property 2.1, the MSUs of each LoRA can be permuted arbitrarily without affecting the functionality of the LoRA module. However, misalignment of MSU parameters when merging LoRAs can result in parameter interference. To investigate the impact of parameter misalignment on model performance, we conducted a controlled experiment using the Llama-2-7b model, training LoRAs on different tasks. For the parameters $A$ and $B$ of a task, we randomly

Table 2: **Parameter interference due to knowledge conflict.** "Tuning MSU" indicates the performance after tuning the added MSU for each task. "Avg MSU" denotes the performance achieved by directly merging these task-specific MSUs. "Concat MSU" represents the performance after concatenating these task-specific MSUs.

| Task | Tuning MSU | Avg MSU | Concat MSU |
|---|---|---|---|
| MNLI | 86.17 | 46.24 (46.35%↓) | 81.36 (5.58%↓) |
| MRPC | 87.25 | 64.75 (25.78%↓) | 81.25 (6.88%↓) |
| Overall | 86.71 | 55.49 (36.06%↓) | 81.31 (6.23%↓) |

generated a permutation matrix $P$ and adjusted the parameters to $A' = (A + PA)/2$ and $B' = (B + BP^T)/2$. *This adjustment simulates the merging of two identical LoRAs with misaligned parameters.* The results, presented in Tab.1, indicate that parameter misalignment can lead

to a decline in model performance, with some tasks experiencing significant performance degradation. Therefore, ideal merging entails aligning MSUs during LoRA merging to mitigate parameter interference. Appendix I shows a more in-depth discussion and analysis.

Another source of parameter interference stems from ***knowledge conflict*** during LoRA merging. As depicted on the right side of Fig.2, knowledge conflict occurs when the merged LoRA lacks sufficient parameter space to encapsulate the comprehensive knowledge. This deficiency forces the merging of task-specific MSUs, resulting in parameter interference. To investigate the impact of knowledge conflict during LoRA merging, we conducted an experiment to demonstrate the performance degradation resulting from merging task-specific MSUs. With a base LoRA trained on the CoLA task, we adapted this LoRA for two new tasks (MNLI and MRPC) by appending an additional MSU to create two separate task-specific LoRAs. Throughout the training process for the new tasks, only the newly introduced MSU for each task was trainable. In this way, the only difference between the LoRAs for MNLI and MRPC was the unique MSU added for each, which encapsulated distinct semantic information tailored to each task. *This setup was designed to create two task-specific LoRAs that differed only in one MSU, allowing us to observe parameter interference when merging these task-specific MSUs.* The results, depicted in Tab.2, demonstrated that averaging the task-specific MSUs from the two LoRAs significantly reduced performance on each task. In contrast, maintaining these task-specific MSUs through concatenation preserved the capabilities specific to each original task. This suggests that ideal merging should maintain task-specific MSUs during LoRA merging to prevent knowledge conflict and effectively resolve parameter interference.

# 3 METHODOLOGY

## 3.1 LoRA-LEGO FRAMEWORK

Based on the motivation that MSUs are the building blocks of LoRA, we can disassemble and reassemble LoRA like playing with LEGO. Here, we propose a flexible and effective method called LoRA-LEGO as shown in Fig.3. This framework is structured around three main procedures: MSU Grouping, MSU Clustering, and LoRA Reconstruction. These steps collectively facilitate the integration of diverse MSUs into a cohesive LoRA, alleviating the parameter interference while LoRA merging.

**MSU Grouping.** The initial stage of merging $p$ LoRAs begins by disassembling each LoRA into various MSUs and grouping all the MSUs from each LoRA together. Let $\{\boldsymbol{A}_i, \boldsymbol{B}_i\}_{i=1}^{p}$ represent the LoRA parameters of a layer with rank $r_i$. Each LoRA module $\boldsymbol{A}_j, \boldsymbol{B}_j$ contains $r_j$ MSUs, denoted by $\{\boldsymbol{s}_{j1}, \boldsymbol{s}_{j2}, \ldots, \boldsymbol{s}_{jr_j}\}$, where $\boldsymbol{s}_{jl} = [\boldsymbol{a}_{jl}, \boldsymbol{b}_{jl}]$ with $\boldsymbol{a}_{jl} = \boldsymbol{A}_j[:, l]$ and $\boldsymbol{b}_{jl} = \boldsymbol{B}_j[l, :]^T$. The MSU pool $\Phi$, which includes MSUs from all the LoRAs to be merged, is constructed as $\Phi = \bigcup_{j=1}^{k} \{\boldsymbol{s}_{j1}, \boldsymbol{s}_{j2}, \ldots, \boldsymbol{s}_{jr_j}\}$.

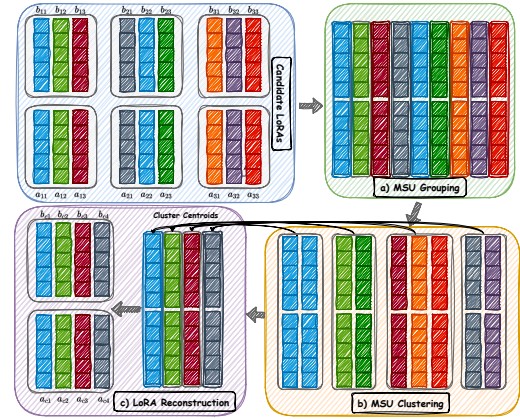

Figure 3: **The LoRA-LEGO framework** merges candidate LoRAs in a manner akin to playing with LEGO by: a) first disassembling LoRAs into multiple MSUs and grouping them into an MSU pool; b) performing MSU clustering to merge similar MSUs; c) reconstructing the merged LoRA from the centroid MSUs to form a cohesive LoRA.

**MSU Clustering.** After grouping the MSUs from different LoRAs, the next step involves regrouping these MSUs into clusters based on their similarities. With the MSU pool $\Phi$, we employed K-means Kanungo et al. (2002) to partition these MSUs into $k$ clusters $\{\mathbb{C}_1, \mathbb{C}_2, \ldots, \mathbb{C}_k\}$ in which each MSU is assigned to the cluster closest to it. This process is described by the following optimization problem:

$$\underset{\mathbb{C}}{\text{minimize}} \quad \sum_{i=1}^{k} \sum_{\boldsymbol{s} \in \mathbb{C}_i} \|\boldsymbol{s} - \boldsymbol{\mu}_i\|^2, \tag{2}$$

where $\boldsymbol{\mu}_i$ is the centroid of cluster $\mathbb{C}_i$.

**LoRA Reconstruction.** Following the MSU clustering, we rearrange the MSUs into $k$ clusters based on their similarity. The centroids of these clusters, denoted by $\boldsymbol{\mu}_1, \boldsymbol{\mu}_2, \ldots, \boldsymbol{\mu}_k$, are calculated as the average of the MSUs within each cluster. These centroids represent aggregated parameters across the MSUs, encapsulating the generalized semantic information most representative of each cluster. Aggregating within each cluster minimizes information loss compared to directly merging different LoRAs, as the MSUs within a cluster are more similar to each other.

Using these $k$ centroids, we can reconstruct a new LoRA module. Each centroid $\boldsymbol{\mu}_i$ contributes to a single rank in the merged model, thus the new LoRA model has a rank $k$, where $1 \leq k \leq \sum_{j=1}^{p} r_j$. The new merged LoRA model is formed by constructing new projection matrices $\boldsymbol{A}'$ and $\boldsymbol{B}'$ from the centroids:

$$\boldsymbol{A}' = \begin{bmatrix} \boldsymbol{a}_1 \\ \boldsymbol{a}_2 \\ \ldots \\ \boldsymbol{a}_k \end{bmatrix}, \quad \boldsymbol{B}' = \begin{bmatrix} \boldsymbol{b}_1^T & \boldsymbol{b}_2^T & \ldots & \boldsymbol{b}_k^T \end{bmatrix}, \tag{3}$$

where $\boldsymbol{a}_i$ and $\boldsymbol{b}_i$ are extracted from each centroid $\boldsymbol{\mu}_i = [\boldsymbol{a}_i, \boldsymbol{b}_i]$ as per the MSU definition. The reconstructed LoRA module $\{\boldsymbol{A}', \boldsymbol{B}'\}$ addresses parameter interference by aligning MSUs based on their similarity before merging, achieving a flexible rank that encapsulates comprehensive knowledge across various tasks. An interesting point is that our method sits between model merging, which fuses multiple identical models into a singular model, and model ensemble, which takes the average of outputs from different modules, achieving a balance between performance and computational efficiency. We provide a detailed discussion of how our method relates to model merging and model ensemble in Appendix B.

## 3.2 Optimal scale of Merged LoRA

Given that the rank of the merged LoRA from LoRA-LEGO can range from 1 to $\sum_{j=1}^{p} r_j$, the scale of LoRA's output could vary significantly, thereby impacting the performance. We identified two key factors that determine the scale of the output.

**Norm Decay After LoRA Merging.** As shown in Fig.4, we examine the norms of the parameters after merging (i.e., the centroids of each cluster) compared to the average norms of the parameters within each cluster before merging. We observed that after merging, the parameter norms significantly decrease, potentially affecting the output scale of the LoRA module, since the parameter norm influences the magnitude of the output. This phenomenon can be explained by the **triangle inequality** Klement et al. (2013), which states that for any vectors $\boldsymbol{s}_i$, $\|\sum_{i=1}^{p} \boldsymbol{s}_i\| \leq \sum_{i=1}^{p} \|\boldsymbol{s}_i\|$. When computing the centroid $\boldsymbol{\mu} = \frac{1}{p} \sum_{i=1}^{p} \boldsymbol{s}_i$, its norm satisfies:

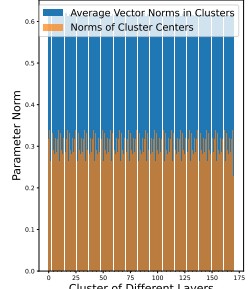

Figure 4: Comparison of *cluster center norm* to *average norm within the cluster*.

$$\|\boldsymbol{\mu}\| = \left\| \frac{1}{p} \sum_{i=1}^{p} \boldsymbol{s}_i \right\| \leq \frac{1}{p} \sum_{i=1}^{p} \|\boldsymbol{s}_i\|.$$

Therefore, the norm of the centroid is less than or equal to the average of the norms of the original vectors, explaining the observed norm decay after merging. The more diverse the vectors within a cluster, the more pronounced this reduction in norm will be. To compensate for the reduced norm after merging, we perform **parameter reweighting** by scaling the centroid to match the average norm of the cluster: $\boldsymbol{\mu}' = \frac{\frac{1}{p} \sum_{i=1}^{p} \|\boldsymbol{s}_i\|}{\|\boldsymbol{\mu}\|} \boldsymbol{\mu}$. In our implementation, we use the *infinity norm* for reweighting to ensure stability and robustness in the results.

**Variance Expansion with Increased LoRA Rank.** Another factor influencing the scale of the LoRA output is the rank of the merged LoRA. We conducted experiments to investigate how the rank of the LoRA affects the output scale by merging seven LoRAs with rank $r = 8$ and varying the rank $k$ of the merged LoRA (which corresponds to the clusters number in LoRA-LEGO). The frequency histograms of outputs from the first layer of the merged LoRA at various ranks, as shown

in Fig.5, indicate that LoRA outputs approximate a normal distribution centered at zero. We observed that as the rank $k$ increases, the variance of the output also increases. To normalize the output variance, similar to the normalization in the self-attention mechanisms Vaswani (2017), we perform **output reweighting** for the merged LoRA by the factor $\frac{\sqrt{r}}{\sqrt{k}}$. The following theorem ensures that this rescaling maintains a consistent variance in the LoRA output.

**Theorem 3.1.** *Let $\boldsymbol{A}_1 \in \mathbb{R}^{p \times r}$ and $\boldsymbol{B}_1 \in \mathbb{R}^{r \times p}$, and $\boldsymbol{A}_2 \in \mathbb{R}^{p \times k}$ and $\boldsymbol{B}_2 \in \mathbb{R}^{k \times p}$, where all elements of these matrices are independently and identically distributed according to the standard normal distribution $\mathcal{N}(0, 1)$. Then, after scaling the product $\boldsymbol{A}_2 \boldsymbol{B}_2$ by the factor $\sqrt{r}/\sqrt{k}$, the variances of the entries of $\boldsymbol{A}_1 \boldsymbol{B}_1$ and the scaled $\boldsymbol{A}_2 \boldsymbol{B}_2$ are equal:*

$$\mathrm{Var}\left(\boldsymbol{A}_1 \boldsymbol{B}_1\right) = \mathrm{Var}\left(\frac{\sqrt{r}}{\sqrt{k}} \boldsymbol{A}_2 \boldsymbol{B}_2\right).$$

The proof of Theorem 3.1 is detailed in the Appendix D. Overall, to ensure that the LoRA output is correctly scaled, we employ two scaling strategies. First, we reweight the parameters to match the average norms of the parameters within each cluster. Second, we rescale the output of the merged LoRA for maintaining variance consistency with the original LoRA. These dual scaling strategies enable LoRA-LEGO to deliver enhanced and more robust performance.

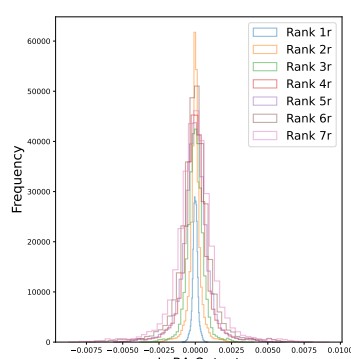

Figure 5: Expansion of variance with increasing rank in merged LoRAs.

## 4 EXPERIMENTS

Given that LoRA merging is essential for many scenarios, we have opted for two settings: Multi-task learning Tang et al. (2024) and Mixed-task settings Zhao et al. (2024b). In these settings, we compared various LoRA composition methods to assess the performance of the proposed LoRA-LEGO approach. We selected Llama2-{7b,13b} as the base model and trained LoRA for each task with hyperparameters $r = 6$ and $\alpha = 12$. The evaluation frameworks for multi-task Learning and mixed-task settings are detailed in the subsequent sections, where we provide a comprehensive analysis.

### 4.1 MULTI-TASK LEARNING

**Experiment Setting.** Multi-task learning aims to merge individually trained LoRAs into a unified model while preserving the performance of each constituent LoRA. Drawing from previous research Tang et al. (2024); Yadav et al. (2024b); Ilharco et al. (2022), we merged seven LoRA models, each fine-tuned on Llama2-{7b,13b}, for in-domain tasks including Cola, Mnli, MRPR, QNLI, GLUE-QQP, RTE, and SST2. We then assessed the performance of the merged LoRA on these in-domain tasks as well as on two additional out-of-domain tasks, SNLI and WNLI, to evaluate its adaptability and generalization capabilities.

**Baseline Methods.** We compared the proposed method with 6 post-hoc training-free LoRA composition methods, including (1) Weight Averaging, (2) Ensemble, (3) Task Arithmetic, (4) Ties-Merging, (5) DARE, and (6) DELLA-Merging. The details of these LoRA composition methods can be found in the Appendix C.

**Main Results.** As shown in Tab. 3, our proposed LoRA-LEGO method significantly outperforms the baseline methods on both IID and OOD tasks. Specifically, the Weight Averaging method suffers from significant performance degradation due to parameter interference during LoRA merging. The Ensemble method encounters issues with parameter redundancy, leading to suboptimal performance and slower inference speeds. Model merging methods such as Task Arithmetic and Ties-Merging perform element-wise fusion and fail to adequately address parameter interference in LoRA, resulting in suboptimal performance during the merging process. Similarly, DARE and DELLA-Merging adopt element-wise parameter merging, which ignores the alignment of the parameter semantic

Table 3: Multi-task performance when merging Llama2-{7b,13b} (LoRA fine-tuned) models on seven seen tasks and two unseen tasks.

| Method | IID Tasks | | | | | | | OOD Tasks | | Average |
|---|---|---|---|---|---|---|---|---|---|---|
| | CoLA | MNLI | MRPC | QNLI | QQP | RTE | SST2 | SNLI | WNLI | |
| *w/ Llama2-7b* | | | | | | | | | | |
| Task LoRA | 61.63 | 77.46 | 68.00 | 77.25 | 75.83 | 52.22 | 75.74 | ╲ | ╲ | ╲ |
| Weight Average | 54.42 | 36.09 | **68.00** | 44.41 | 51.72 | 48.15 | 42.99 | 31.64 | 47.14 | 47.17 |
| Ensemble | **55.67** | 45.89 | 59.25 | 59.84 | 67.38 | 68.89 | 66.44 | 36.73 | 51.43 | 56.84 |
| Task Arithmetic | 55.48 | 42.15 | 54.25 | 58.94 | 66.43 | 67.78 | 59.54 | 34.08 | 54.29 | 54.77 |
| Ties-Mering | 48.65 | 48.81 | 55.50 | 61.79 | 66.75 | 62.59 | 70.69 | 48.45 | **61.43** | 58.30 |
| DARE | 54.62 | 36.16 | 67.75 | 44.41 | 51.83 | 47.78 | 43.45 | 31.64 | 47.14 | 47.20 |
| DELLA-Merging | 55.19 | 36.88 | 53.25 | 56.04 | 65.69 | 60.37 | 57.70 | 31.02 | 51.43 | 51.95 |
| LoRA-LEGO | 55.48 | **55.73** | 66.00 | 62.29 | 71.07 | 71.85 | 73.22 | 51.36 | 52.86 | 62.21 |
| *w/ Llama2-13b* | | | | | | | | | | |
| Task LoRA | 69.04 | 88.23 | 89.25 | 82.33 | 86.29 | 80.74 | 76.44 | ╲ | ╲ | ╲ |
| Weight Average | 45.48 | 46.32 | 67.75 | 46.68 | 47.50 | 62.96 | 46.78 | 42.42 | 42.86 | 49.86 |
| Ensemble | 62.50 | 64.64 | 74.75 | 71.81 | 81.35 | **79.26** | 75.52 | 54.32 | 60.00 | 69.35 |
| Task Arithmetic | **63.17** | 64.41 | 74.50 | 71.59 | 80.84 | 78.15 | 75.40 | 54.16 | 58.57 | 69.03 |
| Ties-Mering | 58.56 | 64.71 | **78.75** | **74.27** | 80.71 | 76.67 | 75.40 | 56.02 | 61.43 | 69.61 |
| DARE | 45.00 | 46.34 | 67.75 | 46.74 | 47.32 | 63.33 | 46.90 | 42.55 | 44.29 | 50.02 |
| DELLA-Merging | 62.21 | 62.45 | 71.25 | 69.05 | 76.20 | 78.52 | 75.40 | 49.86 | 58.57 | 67.06 |
| LoRA-LEGO | 59.42 | **65.40** | 75.50 | 72.29 | **82.51** | 78.52 | **75.98** | 58.54 | 64.29 | 70.27 |

Figure 6: LoRA pruning performance over seven datasets.

space, leading to parameter interference and resulting in suboptimal model merging performance. In contrast, our proposed LoRA-LEGO effectively alleviates parameter misalignment and knowledge conflict through flexible MSU clustering, thereby achieving superior performance compared to other methods. In Appendix E, we demonstrate that the proposed LoRA-LEGO approach can effectively merge heterogeneous LoRAs, exceeding the capabilities of previous model merging methods. In Appendix F, we explore the results of model merging on more diverse tasks and find that task diversity introduces greater challenges for LoRA merging. In such scenarios, our method significantly outperforms other baseline methods.

**Performance on LoRA Pruning.** Our method also functions as a LoRA parameter pruning approach. For a single LoRA with rank $r$, LoRA-LEGO allows for selecting $k < r$, effectively reducing the rank to $k$ and pruning the model. As illustrated in Fig. 6, we evaluate the performance of a single LoRA model after retaining various proportions of its parameters. LoRA-LEGO efficiently compresses model parameters: retaining just 33% of the parameters preserves 79% of the original model's capabilities while keeping 50% maintains 92% of the performance. This offers new insights into strategies for compressing model parameters, especially those of LoRA.

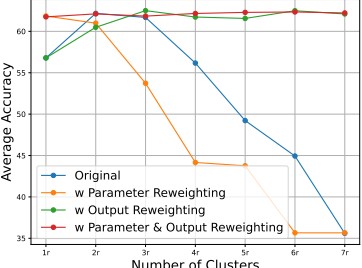

Figure 7: Ablation on scaling strategies.

**Ablation of Scaling Strategies.** We evaluate the effectiveness of two scaling strategies for the merged LoRA by varying the number of clusters for LoRA-LEGO, noting that the cluster number corresponds to the rank of the merged LoRA. As illustrated in Fig.7, the original computation of LoRA experiences significant performance degradation with increasing rank of the merged LoRA, primarily due to the expansion of variance associated with the increased rank. Additionally, when the rank of the merged LoRA is relatively low, its performance does not reach its optimum due to the degradation of parameter norms. We also present the performance of each scaling strategy and their combination. Applying **parameter reweighting** can significantly enhance the performance of the merged LoRA when the rank is relatively low; specifically, the performance of a merged LoRA at rank $1r$ improves by 5%. However, as the rank increases, eliminating norm decay more severely exposes variance expansion because norm decay can alleviate this phenomenon, leading to greater performance degradation. Stabilizing the variance by **output reweighting** significantly increases

Table 4: The average performance of each task cluster. The performance of perfectly selected corresponding LoRA for each sample is colored in gray. We have bolded the best performance of each task and underlined the best performance in the "OOD" setting.

| Task | Perfect Selection | LoRA-LEGO | | Selection | | Weight Average | | Ensemble | | Tie-Merging | |
|---|---|---|---|---|---|---|---|---|---|---|---|
| | | IID | OOD | IID | OOD | IID | OOD | IID | OOD | IID | OOD |
| *w/ Llama2-7b* | | | | | | | | | | | |
| Struct to Text Rouge-1 | 59.1 | 49.6 | 50.5 | **56.8** | 45.2 | 44.5 | 41.0 | 51.2 | 45.3 | 45.4 | 49.4 |
| Struct to Text Rouge-2 | 36.1 | 25.7 | 26.6 | **33.6** | 23.2 | 22.6 | 20.2 | 26.3 | 22.9 | 23.9 | 27.4 |
| Struct to Text Rouge-l | 48.6 | 39.5 | 39.9 | **46.4** | 35.3 | 34.5 | 31.7 | 41.0 | 35.5 | 36.0 | 39.3 |
| Translation BLEU | 13.1 | **12.9** | 12.4 | 12.8 | 12.0 | 12.2 | 12.3 | 12.8 | 12.2 | 14.0 | 13.9 |
| COMMONSENSE | 62.5 | **60.0** | 60.5 | 55.5 | 46.0 | 51.0 | 48.0 | 61.5 | 50.0 | 55.0 | 59.5 |
| SENTIMENT | 90.0 | **90.0** | 91.5 | 89.5 | 89.0 | 79.0 | 78.5 | 89.5 | 90.5 | 82.0 | 81.5 |
| READING Comp. | 67.3 | **54.3** | 55.7 | 51.7 | 40.3 | 47.3 | 45.0 | 51.3 | 47.3 | 46.3 | 56.3 |
| CLOSE-BOOK QA | 45.0 | 47.0 | 48.5 | 40.0 | 43.0 | 41.0 | 37.5 | 45.0 | 48.5 | **48.0** | 53.5 |
| COREFERENCE | 52.0 | **62.0** | 60.0 | 50.0 | 46.0 | 47.0 | 53.0 | 63.0 | 49.0 | 32.0 | 47.0 |
| READ. COOMP. W/ COM | 69.0 | 66.0 | 65.0 | **69.0** | 30.0 | 35.0 | 19.0 | 46.0 | 40.0 | 37.0 | 64.0 |
| PARAPHRASE | 65.5 | **58.0** | 60.0 | **58.0** | 45.5 | 45.5 | 44.0 | 56.5 | 45.5 | 18.0 | 38.5 |
| NLI | 72.3 | **71.3** | 66.4 | 70.0 | 60.6 | 51.4 | 53.8 | 67.9 | 64.3 | 65.6 | 49.4 |
| Overall | 55.4 | **51.4** | 51.0 | 51.2 | 43.0 | 41.6 | 40.2 | 49.8 | 45.6 | 43.2 | 45.8 |
| *w/ Llama2-13b* | | | | | | | | | | | |
| Struct to Text$_{Rouge-1}$ | 61.0 | 54.2 | 46.0 | **58.0** | 44.6 | 48.2 | 45.1 | 52.9 | 46.9 | 50.8 | 50.9 |
| Struct to Text$_{Rouge-2}$ | 37.7 | 29.3 | 24.0 | **34.9** | 22.8 | 26.0 | 23.5 | 29.1 | 24.6 | 26.2 | 26.1 |
| Struct to Text$_{Rouge-l}$ | 50.5 | 43.9 | 36.4 | **47.6** | 34.8 | 38.4 | 35.9 | 42.9 | 36.9 | 41.0 | 40.9 |
| Translation$_{BLEU}$ | 12.9 | **14.7** | 14.5 | 12.9 | 12.7 | 14.6 | 14.1 | 14.6 | 14.1 | 11.2 | 11.3 |
| COMMONSENSE | 69.5 | **69.0** | 68.5 | 59.0 | 47.5 | 61.0 | 56.0 | 64.0 | 60.5 | 58.0 | 57.5 |
| SENTIMENT | 90.0 | 91.0 | 90.0 | 90.5 | 91.0 | 87.0 | 83.5 | **91.5** | 91.5 | 91.5 | 91.5 |
| READING Comp. | 76.0 | **62.7** | 53.0 | 60.3 | 48.0 | 56.7 | 49.3 | 60.3 | 51.3 | 54.3 | 54.3 |
| CLOSE-BOOK QA | 64.0 | **63.0** | 58.0 | 60.0 | 53.0 | 62.0 | 58.0 | 63.0 | 61.0 | 41.5 | 42.0 |
| COREFERENCE | 74.0 | **77.0** | 62.0 | 75.0 | 65.0 | 55.0 | 59.0 | 76.0 | 64.0 | 63.0 | 63.0 |
| READ. COOMP. W/ COM | 82.0 | 76.0 | 54.0 | 80.0 | 33.0 | 57.0 | 49.0 | 78.0 | 58.0 | 65.0 | 66.0 |
| PARAPHRASE | 77.5 | 67.5 | 58.5 | 68.0 | 52.5 | 55.5 | 45.5 | 71.0 | 55.5 | 61.0 | 62.5 |
| NLI | 82.4 | **78.9** | 76.3 | **78.9** | 70.2 | 69.8 | 66.4 | 78.1 | 75.7 | 65.7 | 65.7 |
| Overall | 62.4 | **58.2** | 52.8 | 57.8 | 47.7 | 51.6 | 47.8 | 57.6 | 52.3 | 50.1 | 50.3 |

performance when the rank is high, although it remains suboptimal due to the decrease of parameter norms. Combining these two scaling strategies yields the best results, demonstrating stable and improved performance across varying ranks of the merged LoRA. Notably, our approach's only hyperparameter is the cluster number $k$. After applying the dual rescaling strategy, our method's performance becomes highly stable, indicating that it is very robust regarding hyperparameter selection; therefore, we use $k = 2r$ as the default setting.

**Merging Different Number of Tasks.** We investigated the average performance of the model when merging LoRAs with different numbers of tasks. To better assess the influence of task quantity on our method, we normalized the performance of each task by dividing it by the performance of its respective single-task LoRA and then calculated the mean of these normalized scores. From Fig.8, it is evident that as the number of merging tasks increases, there is a general decline in the performance of the merged LoRAs. Specifically, direct averaging experiences a steep performance drop due to parameter interference. The Ensemble method also sees a decrease in performance, attributed to parameter redundancy and inconsistencies in the output space. Ties-merging, failing to resolve parameter interference and reliant on hyperparameter selection fully, does not reach optimal performance. LoRA-LEGO, which flexibly addresses parameter interference, experiences a lesser decline in performance with an increasing number of tasks, thereby outperforming the baseline model. In Appendix G, we discuss the scalability of various model merging methods and showcase their performance in merging LoRAs for 20 diverse tasks. The results clearly highlight the efficiency and effectiveness of our approach.

## 4.2 MIXED-TASK EVALUATION

**Evaluation Setting.** Recent studies Zhao et al. (2024b) have proposed the creation of a LoRA pool from which relevant LoRAs are retrieved for each input to facilitate LoRA composition. We adopt the same setting and construct a LoRA pool for 48 tasks from flan-v2, grouped into 10 task clusters. The evaluation set is constructed by randomly choosing 50 samples from each test set. These samples are then mixed and shuffled to form a unified dataset comprising 2400 data points.

Figure 8: Average performance varying the number of merged tasks.

Adopting the LoraRetriever approach Zhao et al. (2024b), we initially retrieve the top-3 LoRAs based on the sentence embedding similarities between each input sample and LoRA's few-shot samples. Following this, we engage in LoRA composition and evaluate various strategies. This analysis underscores the versatility and superior performance of LoRA-LEGO in handling more complex scenarios.

**Baseline Methods.** For all methods, we employ a consistent evaluation pipeline. For each instance in the evaluation set, we initially retrieve the top-3 LoRA, followed by the composition of LoRA. We compared the following LoRA composition methods: (1) Weight Average, (2) Ensemble, (3) Selection (using the top-1 retrieved LoRA), and (4) Ties-Merging.

**Main Results.** Previous research Zhao et al. (2024b) has shown that using a retriever to identify LoRA tasks tailored to various inputs is more efficient and effective in personalized service settings. Consequently, we concentrate on how multiple LoRAs can be integrated effectively through LoRA merging after retrieving the top-k LoRAs for each input. We assess the performance of LoRA composition methods in both IID and OOD contexts. "IID" performance refers to scenarios where all LoRAs are accessible to the retriever. "OOD" performance, however, involves masking the LoRA associated with the specific task of each test sample during retrieval, preventing any sample from accessing its ideal LoRA. This approach allows us to evaluate the cross-task generalization capabilities of the LoRA composition methods. Tab.4 demonstrates that LoRA-LEGO surpasses other composition methods in both IID and OOD scenarios by fully eliminating parameter interference. In contrast, baseline LoRA composition methods experience performance degradation due to their inability to completely mitigate parameter interference. Specifically, in IID scenarios, the Selection method excels because the Retriever can choose the most appropriate LoRA from closely related tasks for inference. Building on this, LoRA-LEGO further enhances performance by leveraging the transfer capabilities between different tasks, thereby achieving better results. For OOD scenarios, both Ties-Merging and Ensemble show good performance by harnessing knowledge from a wide array of relevant tasks to tackle OOD tasks. LoRA-LEGO, however, outperforms these methods by effectively addressing parameter interference, allowing for a more comprehensive utilization of diverse LoRA capabilities and achieving superior results in OOD setting.

## 5 RELATED WORK

Many works have discussed how to obtain a comprehensive model through model merging from various perspectives. Some works discuss how to find a set of low-loss paths in the parameter space for model parameter interpolation from the perspective of linear mode connectivity Ainsworth et al. (2022); Entezari et al. (2021). From a similar perspective, we further utilized properties of MSUs, employing clustering algorithms to provide a flexible solution for enhancing the parameter connectivity during LoRA merging. Additionally, many works attempt to coordinate models trained in a decentralized and separated manner through model merging, utilizing their knowledge transfer capabilities to obtain a model with comprehensive abilities Tang et al. (2024); Don-Yehiya et al. (2022); Yadav et al. (2024b); Matena & Raffel (2022); Jin et al. (2022); Yang et al. (2023a); Deep et al. (2024); Yu et al. (2024). Recently, with the rise of large language models, more and more works have focused on how to use model aggregation, especially the aggregation of LoRA Chronopoulou et al. (2023); Huang et al. (2023); Zhao et al. (2024b); Wang et al. (2024), to strategically utilize models adapted to multiple domains. These efforts often overlook the parameter interference that occurs during LoRA merging, and some of them require extensive additional training or adaptation. This leads to suboptimal performance in such scenarios or restricts their applicability. More detailed related work can be found in Appendix H.

## 6 CONCLUSION

In this paper, we address the challenge of merging multiple task-specific LoRAs into a unified model. We identify parameter interference, caused by misalignment and knowledge conflict, as the main obstacle. Our analysis reveals key LoRA properties: (1) Each rank operates as a minimal semantic unit (MSU); (2) MSUs exhibit permutation invariance; (3) MSUs can be concatenated to form a comprehensive LoRA. Building on these insights, we propose LoRA-LEGO, which clusters MSUs from target LoRAs and uses cluster centroids to create a merged LoRA. Extensive experiments validate our approach's effectiveness. Potential future work includes exploring alternative distance metrics for LoRA-LEGO, such as optimal transport, to more effectively capture parameter similarities beyond the standard Euclidean distance. Additionally, further modularization of LoRA could improve various applications, such as federated learning. We believe these advancements have the potential to significantly benefit a broad range of fields and applications.

ACKNOWLEDGEMENTS

This work was supported by the National Natural Science Foundation of China (62441605, 62441617).

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

## A    DIFFERENCE BETWEEN LoAR MERGING SETTING AND MODEL MERGING SETTING

Previous work on model merging primarily focused on integrating separately trained models to form a comprehensive system. These methods typically involve reloading LoRA parameters into the original model before merging, which introduces additional overhead by necessitating the reconstruction of a corresponding LLM for each LoRA. In many cases, the goal of LoRA merging is to create a new LoRA that consolidates the capabilities of all involved LoRAs for simplified task-specific usage. In contrast, the LoRA merging setting presented in this paper bypasses the LoRA reload step; it directly merges the LoRA parameters to construct a unified LoRA with comprehensive capabilities.

## B    CONNECTION WITH VANILLA LoRA COMPOSITION METHODS

The vanilla LoRA composition can be categorised into two types of training-free methods: the model ensembling and model merging Tang et al. (2024). The **ensemble** strategy involves aggregating the outputs of each submodule within the assembled LoRAs. Let us denote $\mathcal{A} = \{\boldsymbol{A}_1, \boldsymbol{A}_2, \ldots, \boldsymbol{A}_n\}$ and $\mathcal{B} = \{\boldsymbol{B}_1, \boldsymbol{B}_2, \ldots, \boldsymbol{B}_n\}$ as the sets representing submodules within $n$ LoRAs. For an input $\boldsymbol{x}_i$, the output derived from the ensemble of LoRAs can be expressed as $\boldsymbol{x}_i' = \frac{1}{n} \sum_{j=1}^{n} \boldsymbol{B}_j \boldsymbol{A}_j \boldsymbol{x}_i$, where $\boldsymbol{x}_i'$ denotes the output. The performance of the ensemble of LoRAs tends to be more stable, but it incurs additional computational overhead. In contrast to the ensemble method, **model merging** presents an alternative composition strategy. A typical strategy involves employing an element-wise fusion of these parameters, represented as $\boldsymbol{A}' = \frac{1}{n} \sum_{j=1}^{n} \boldsymbol{A}_j$ and $\boldsymbol{B}' = \frac{1}{n} \sum_{j=1}^{n} \boldsymbol{B}_j$. This formulation allows the merged parameters to function similarly to a single LoRA. However, directly merging parameters can lead to performance degradation due to parameter interference.

Our proposed LoRA-LEGO method serves as a bridge between the two strategies, ensuring an optimal balance between computational efficiency and performance. By selectively aligning and fusing MSUs based on their semantic similarity, LoRA-LEGO effectively condenses the most relevant semantic features into fewer clusters. This process allows for the merging of parameters within each cluster, reducing the overall parameter count in a manner similar to the model merging method. By adjusting the number of clusters, LoRA-LEGO can accommodate more parameters for inference,

much like the ensemble method. In this way, our method leverages the strengths of both methodologies, ultimately enhancing model performance and inference efficiency.

## C DETAILS OF BASELINE METHODS

We compare our method with the following baseline:

1. **Weight Averaging**. This approach averages the parameters across different instances of LoRA, resulting in a new composite LoRA defined as $\boldsymbol{A}' = \frac{1}{n}\sum_{i=1}^{n}\boldsymbol{A}_i$ and $\boldsymbol{B}' = \frac{1}{n}\sum_{i=1}^{n}\boldsymbol{B}_i$, where $\boldsymbol{A}_i$ and $\boldsymbol{B}_i$ represent the parameters from the $i$-th instance of the original LoRA models, and $n$ is the number of models being averaged.

2. **Ensemble**. This method averages the outputs from each LoRA, simultaneously activating multiple LoRAs to compose a combined output. The specific calculation for the mixed output is defined as $\boldsymbol{x}' = \frac{1}{n}\sum_{i=1}^{n}\boldsymbol{B}_j\boldsymbol{A}_j\boldsymbol{x}_i$.

3. **Task Arithmetic**. This method is akin to weight averaging, but it differentiates by using weights derived from a hyper-parameter search to merge models. The calculations for this composite are $\boldsymbol{A}' = p\sum_{i=1}^{n}\boldsymbol{A}_i$ and $\boldsymbol{B}' = p\sum_{i=1}^{n}\boldsymbol{B}_i$, where $p$ represents the hyper-parameter that scales the contributions of each model.

4. **Ties-Merging**. This method aims to resolve element-wise parameter interference by initially trimming the redundant parameters, retaining only the top-k% of values based on their magnitude. It then selects the sign vector for the merged model and finally performs a disjoint mean operation. Ties-Merging posits that the primary source of parameter interference arises from inconsistencies in the values of merged parameters, while potentially overlooking issues related to misalignment and knowledge conflict.

5. DARE. This method randomly drops the parameters and then rescales the remaining values to maintain the model's performance on the target tasks.

6. DELLA-Merging. This method pruning the parameters taking into account their magnitude, then elects the parameters for merging to reduce parameter interference.

## D OPTIMAL SCALE OF MERGED LoRA

**Theorem D.1.** *Let $\boldsymbol{A}_1 \in \mathbb{R}^{p \times r}$ and $\boldsymbol{B}_1 \in \mathbb{R}^{r \times p}$, and $\boldsymbol{A}_2 \in \mathbb{R}^{p \times k}$ and $\boldsymbol{B}_2 \in \mathbb{R}^{k \times p}$, where all elements of these matrices are independently and identically distributed according to the standard normal distribution $\mathcal{N}(0, 1)$. Then, after scaling the product $\boldsymbol{A}_2 \boldsymbol{B}_2$ by the factor $\sqrt{r}/\sqrt{k}$, the variances of the entries of $\boldsymbol{A}_1 \boldsymbol{B}_1$ and the scaled $\boldsymbol{A}_2 \boldsymbol{B}_2$ are equal:*

$$\mathrm{Var}\left(\boldsymbol{A}_1 \boldsymbol{B}_1\right) = \mathrm{Var}\left(\frac{\sqrt{r}}{\sqrt{k}}\boldsymbol{A}_2 \boldsymbol{B}_2\right).$$

*Proof.* To compute the variance of the entries of the matrices $\boldsymbol{A}_1 \boldsymbol{B}_1$ and $\frac{\sqrt{r}}{\sqrt{k}}\boldsymbol{A}_2 \boldsymbol{B}_2$, we examine each entry individually.

For $\boldsymbol{A}_1 \boldsymbol{B}_1$, each entry is calculated as:

$$(\boldsymbol{A}_1 \boldsymbol{B}_1)_{ij} = \sum_{l=1}^{r}(\boldsymbol{A}_1)_{il}(\boldsymbol{B}_1)_{lj}.$$

Since $(\boldsymbol{A}_1)_{il}$ and $(\boldsymbol{B}_1)_{lj}$ are independent and follow $\mathcal{N}(0, 1)$, their product has mean zero and variance one:

$$\mathbb{E}\left[(\boldsymbol{A}_1)_{il}(\boldsymbol{B}_1)_{lj}\right] = 0, \quad \mathrm{Var}\left((\boldsymbol{A}_1)_{il}(\boldsymbol{B}_1)_{lj}\right) = 1.$$

The terms $(\boldsymbol{A}_1)_{il}(\boldsymbol{B}_1)_{lj}$ are independent for different $l$, so the variance of $(\boldsymbol{A}_1 \boldsymbol{B}_1)_{ij}$ is:

$$\mathrm{Var}\left((\boldsymbol{A}_1 \boldsymbol{B}_1)_{ij}\right) = \sum_{l=1}^{r}\mathrm{Var}\left((\boldsymbol{A}_1)_{il}(\boldsymbol{B}_1)_{lj}\right) = r \times 1 = r.$$

Table 5: Multi-task performance when merging heterogeneous LoRAs on seven seen tasks and two unseen tasks.

| Method | IID Tasks | | | | | | | OOD Tasks | | Average |
| | CoLA | MNLI | MRPC | QNLI | QQP | RTE | SST2 | SNLI | WNLI | |
| --- | --- | --- | --- | --- | --- | --- | --- | --- | --- | --- |
| *w/ Llama2-7b* | | | | | | | | | | |
| Task LoRA | 61.63 | 77.46 | 68.00 | 82.69 | 75.83 | 77.04 | 77.47 | ╲ | ╲ | ╲ |
| Weight Average | ╲ | ╲ | ╲ | ╲ | ╲ | ╲ | ╲ | ╲ | ╲ | ╲ |
| Task Arithmetic | ╲ | ╲ | ╲ | ╲ | ╲ | ╲ | ╲ | ╲ | ╲ | ╲ |
| Ties-Mering | ╲ | ╲ | ╲ | ╲ | ╲ | ╲ | ╲ | ╲ | ╲ | ╲ |
| Ensemble | **56.06** | 55.84 | **69.75** | 64.91 | **74.85** | **74.44** | 70.92 | 46.19 | **52.86** | 62.87 |
| LoRA-LEGO | 55.10 | **60.67** | 69.25 | **67.29** | 65.61 | 67.04 | **74.83** | **57.82** | **52.86** | **63.39** |

Similarly, for $\boldsymbol{A}_2\boldsymbol{B}_2$, each entry is:

$$(\boldsymbol{A}_2\boldsymbol{B}_2)_{ij} = \sum_{l=1}^{k}(\boldsymbol{A}_2)_{il}(\boldsymbol{B}_2)_{lj},$$

and each term $(\boldsymbol{A}_2)_{il}(\boldsymbol{B}_2)_{lj}$ has variance one. Therefore, the variance of $(\boldsymbol{A}_2\boldsymbol{B}_2)_{ij}$ is:

$$\text{Var}\left((\boldsymbol{A}_2\boldsymbol{B}_2)_{ij}\right) = \sum_{l=1}^{k}\text{Var}\left((\boldsymbol{A}_2)_{il}(\boldsymbol{B}_2)_{lj}\right) = k \times 1 = k.$$

After scaling $\boldsymbol{A}_2\boldsymbol{B}_2$ by $\sqrt{r}/\sqrt{k}$, the variance becomes:

$$\text{Var}\left(\left(\frac{\sqrt{r}}{\sqrt{k}}\boldsymbol{A}_2\boldsymbol{B}_2\right)_{ij}\right) = \left(\frac{\sqrt{r}}{\sqrt{k}}\right)^2\text{Var}\left((\boldsymbol{A}_2\boldsymbol{B}_2)_{ij}\right) = \left(\frac{r}{k}\right) \times k = r.$$

Thus, the variances of the entries are equal:

$$\text{Var}\left(\boldsymbol{A}_1\boldsymbol{B}_1\right) = \text{Var}\left(\frac{\sqrt{r}}{\sqrt{k}}\boldsymbol{A}_2\boldsymbol{B}_2\right).$$

$\square$

# E  PERFORMANCE ON MERGING HETEROGENEOUS LoRAs

Another advantage of LoRA-LEGO is its ability to merge **heterogeneous LoRAs**, that is, LoRAs with different ranks. To experimentally verify this feature, we retrained LoRAs for the QNLI, RTE, and SST2 tasks with $r = 16$ and $\alpha = 32$, and merged them with LoRAs from other tasks ($r = 8$, $\alpha = 16$) to obtain a new LoRA. Since other model merging methods require the merged LoRAs to have the same architecture, we only compared our method with the Ensemble method. As shown in Tab.5, the results demonstrate that our method can effectively merge heterogeneous LoRAs and achieves better overall performance than the Ensemble method.

# F  PERFORMANCE ON MERGING LoRAs FROM DIVERSE TASKS

To comprehensively evaluate different merging methods' performance on diverse tasks, we conducted experiments with LoRA models trained across a broad spectrum of NLP tasks. These tasks include RTE (natural language inference), CoPA (commonsense reasoning), IMDB reviews (sentiment analysis), MRPC (paraphrase detection), BoolQ (reading comprehension), CosmosQA (reading comprehension with commonsense), and ARC (closed-book question answering). The deliberate selection of these highly diverse tasks enables a thorough assessment of merging methods across different linguistic capabilities and reasoning requirements.

The performance comparison of different LoRA merging methods on these diverse tasks is presented in Tab.6. As expected, the significant semantic disparities between tasks lead to increased parameter interference, causing substantial performance degradation in baseline methods such as ties-merging and DARE. In contrast, LoRA-LEGO's approach of parameter alignment before MSU-based merging proves highly effective, achieving an 11% performance improvement over the strongest baseline,

Table 6: Multi-task performance when merging LoRAs from diverse tasks.

| Method | RTE | CoPA | IMDB | MRPC | BoolQ | CosmosQA | ARC(Easy) | Average |
|---|---|---|---|---|---|---|---|---|
| Weight Average | 47.41 | 51.00 | 5.90 | 56.50 | 60.70 | 20.80 | 12.70 | 36.43 |
| Ensemble | 52.59 | 3.00 | 33.20 | 35.00 | 73.50 | 0.50 | 19.10 | 30.98 |
| Ties-Merging | 38.89 | 23.00 | 0.00 | 0.00 | 10.20 | 8.90 | 1.70 | 11.81 |
| DARE | 47.41 | 53.00 | 6.40 | 57.25 | 60.70 | 21.20 | 12.90 | 36.98 |
| DELLA-Merging | 55.56 | 63.00 | 58.99 | 68.25 | 72.63 | 18.02 | 38.90 | 53.62 |
| LoRA-LEGO | **56.30** | **69.00** | **65.10** | **60.00** | **80.40** | **35.50** | **50.40** | **59.53** |

DELLA-Merging. These results demonstrate that our MSU-level alignment and merging strategy successfully mitigates parameter interference, particularly in challenging scenarios where task diversity amplifies interference effects. The superior performance validates our approach's effectiveness in handling heterogeneous task combinations while maintaining model capabilities.

## G    EXPLORING THE SCALABILITY OF LORA MERGING

To rigorously evaluate the **scalability** of different LoRA merging methods and their capacity to handle increasingly diverse tasks, we designed a comprehensive experiment spanning a broad spectrum of tasks and domains. Our evaluation encompasses the following task categories:

- Natural Language Inference: RTE, CB, SNLI, MNLI (matched and mismatched), WNLI, QNLI
- Commonsense Reasoning: CoPA, PiQA, StoryCloze
- Sentiment Analysis: IMDB, SST-2
- Paraphrase Detection and Similarity: MRPC, QQP, STS-B
- Reading Comprehension with Commonsense: CosmosQA
- Reading Comprehension: BoolQ, RTE
- Closed-Book Question Answering: ARC (easy and challenge sets)

We conducted extensive experiments merging 20 LoRAs from these diverse tasks, with results presented in Tab.7. The findings reveal several critical insights about large-scale, multi-domain LoRA merging: Traditional element-wise merging methods exhibit severe performance degradation, with some merged models failing completely due to intensified parameter interference. The ensemble method, in particular, failed entirely due to irreconcilable conflicts arising from disparate output spaces across multiple LoRA models. While DELLA-Merging achieved the best performance among baselines through its parameter pruning strategy, it still struggled to fully address the fundamental challenges of parameter misalignment and knowledge conflicts inherent in large-scale merging. In contrast, LoRA-LEGO demonstrated robust and superior performance in this challenging scenario. By leveraging MSU-based structuring and integration of LoRA knowledge and parameters, our method effectively resolves parameter interference issues, achieving a substantial 16% improvement over the best baseline. These results convincingly demonstrate LoRA-LEGO's effectiveness in large-scale LoRA merging and its ability to maintain performance even under significant task diversity and scale.

## H    DETAILED RELATED WORK ON MODEL MERGING

**Model Merging Methods.**    Model Merging has garnered significant attention, as it aims to combine multiple task-specific models into a unified model capable of handling diverse tasks Tang et al. (2024); Yang et al. (2024a). Previous work, such as **Weight Averaging** Tang et al. (2024), approached model merging by directly averaging all the parameters. **Task Arithmetic** Zhang et al. (2023); Ilharco et al. (2022) introduced additional scaling factors to determine each model's contribution to the final output. Subsequently, methods like **Fisher Merging** Matena & Raffel (2022) and **RegMean** Jin et al. (2022) incorporated additional labeled data to merge models based on their relative importance. **TIES-Merging** Yadav et al. (2024b) tackled element-wise parameter interference by trimming low-magnitude parameters, resolving sign disagreements, and selectively merging parameters with consistent signs. More recent approaches, such as **DARE** Yu et al. (2024) and **DELLA-Merging** Deep et al. (2024), further optimized TIES-Merging by introducing parameter

Table 7: Multi-task performance when merging LoRAs on 20 tasks.

| Task | Weight Average | Ensemble | Ties-Merging | DARE | DELLA-Merging | LoRA-LEGO |
|------|----------------|----------|--------------|------|---------------|-----------|
| ARC (Easy) | 1.00 | 0.00 | 3.40 | 1.30 | 37.00 | **50.20** |
| ARC (Challenge) | 1.30 | 0.00 | 2.50 | 1.30 | 27.90 | **35.20** |
| CoLA | 0.00 | 0.00 | 0.20 | 0.00 | 54.30 | **54.60** |
| QQP | 0.00 | 0.00 | 0.20 | 0.00 | **60.20** | 57.30 |
| PiQA | 1.60 | 0.00 | 7.10 | 1.20 | 45.10 | **45.60** |
| SST2 | 1.38 | 0.00 | 7.24 | 2.07 | 55.40 | **74.02** |
| CoPA | 18.00 | 0.00 | 3.00 | 21.00 | 59.00 | **70.00** |
| IMDB | 0.00 | 0.00 | 1.80 | 0.00 | 51.20 | **64.50** |
| QNLI | 14.70 | 0.00 | 0.90 | 15.30 | 50.30 | **52.50** |
| StoryCloze | 40.00 | 0.00 | 5.50 | 40.10 | 63.90 | **64.00** |
| BoolQ | 12.80 | 0.00 | 5.80 | 13.00 | 69.20 | **81.80** |
| CosmosQA | 9.30 | 0.00 | 3.90 | 9.70 | 16.40 | **16.50** |
| MNLI (m) | 5.90 | 0.00 | 0.60 | 5.90 | 39.70 | **59.10** |
| MNLI (mm) | 3.60 | 0.00 | 0.30 | 3.10 | 40.10 | **56.20** |
| RTE | 39.63 | 0.00 | 3.70 | 39.63 | 57.78 | **62.22** |
| STSB | 0.20 | 0.00 | 0.40 | 0.20 | 15.70 | **17.80** |
| CB | 12.00 | 0.00 | 4.00 | 8.00 | **74.00** | **74.00** |
| MRPC | 0.25 | 0.00 | 0.75 | 0.25 | 59.00 | **66.75** |
| SNLI | 0.50 | 0.00 | 0.70 | 0.80 | 35.80 | **55.10** |
| WNLI | 5.71 | 0.00 | 0.00 | 8.57 | 48.57 | **57.14** |
| Average | 8.39 | 0.00 | 2.60 | 8.57 | 48.03 | **55.73** |

pruning and rescaling techniques. However, these methods mostly rely on element-wise parameter merging and fail to address the alignment of parameters, which leads to significant **parameter interference** in more complex scenarios.

**Application of LoRA Merging.** LoRA merging can be applied in various scenarios. For instance, in multi-task learning Tang et al. (2024); Don-Yehiya et al. (2022), models adapt to different domains in a decentralized manner using LoRA, subsequently acquiring multi-task capabilities through merging. In mixed-task scenarios Zhao et al. (2024b;a), LoRAs from diverse domain tasks are uploaded to a centralized service platform, where the service retrieves and composes LoRAs to deliver personalized services based on downstream requests. In federated learning Chen et al. (2023); Zhang et al. (2024), edge devices train LoRAs on private data and upload them to a central server for merging and distribution, enabling iterative optimization through this process.

# I   EXPLORATION OF AVERAGE MSU DISTANCES WITHIN LORAS

Our experiments in Tab.1 revealed an interesting phenomenon: while merging a LoRA with its permuted version typically leads to performance degradation due to MSU misalignment and parameter interference, certain tasks like RTE and MRPC showed slight performance improvements. We hypothesize that this unexpected benefit stems from these tasks' relative simplicity, where internal MSUs within their LoRA models maintain closer alignment. In such cases, permutation merging may actually serve as a beneficial regularization mechanism, potentially reducing overfitting.

To validate this hypothesis, we analyzed the average Euclidean distances between MSUs in each layer across all LoRAs, as illustrated in Fig.I. The analysis reveals that MRPC and RTE consistently exhibit smaller inter-MSU distances, indicating stronger internal parameter connectivity and potential parameter redundancy. This structural characteristic explains why permutation merging acts as an effective regularization mechanism for these tasks: when MSUs are closely connected, the smoothing effect of permutation merging helps reduce overfitting, leading to performance improvements. Conversely, for tasks with larger MSU distances indicating weaker connectivity, permutation merging disrupts the parameter structure, resulting in significant performance deterioration.

These findings underscore a crucial insight for LoRA merging: the effectiveness of merging strategies strongly depends on MSU connectivity patterns. Our clustering-based approach capitalizes on this insight by specifically targeting and merging closely aligned MSUs, thereby minimizing parameter interference and optimizing merger outcomes.

# J   EXPLORATION OF SEMANTIC RELATIONSHIPS IN MSUS ACROSS TASKS

To investigate whether MSUs capture meaningful structural semantic information, we conducted a comprehensive analysis using 48 LoRA models, each trained independently on different datasets from Flan-v2 and categorized into 10 distinct task clusters. Our analysis focuses on visualizing the

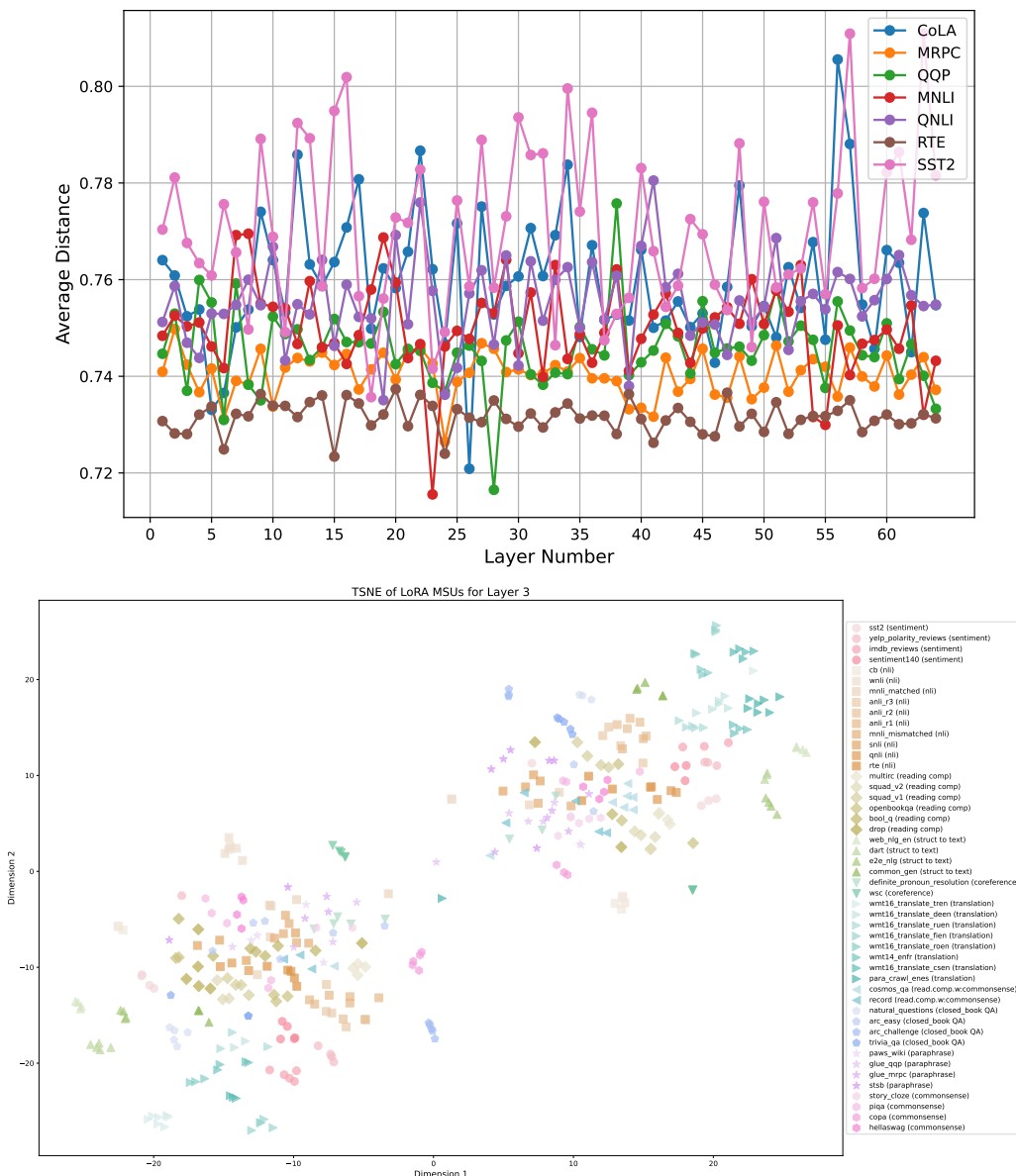

Figure 10: Average Euclidean distance among MSUs of different layers within each LoRA.

semantic relationships between MSUs by examining their distributions in the first layer parameters of these LoRAs.

To uncover potential patterns in MSU organization, we employed t-SNE to project MSUs from the initial layers of these models into a unified visualization space. This approach allows us to examine how MSUs are distributed across different tasks and domains, potentially revealing underlying semantic relationships and organizational principles.

The visualization results, presented in Fig.10, reveal two significant patterns: First, MSUs within individual LoRA tend to form distinct clusters, reflecting their shared training on specific data distributions. Second, and more notably, MSUs from LoRAs within the same domain exhibit spatial proximity in the t-SNE space, suggesting the capture of domain-specific semantic features. These findings have important implications for LoRA merging: they demonstrate that MSU similarity is a crucial consideration in the merging process, as clustering and merging similar MSUs can minimize parameter interference while preserving semantic integrity. This observation supports our approach of MSU-based alignment in LoRA merging, showing how it helps maintain semantic coherence while reducing conflicts during merging.

