# OpenReview forum: "Merging LoRAs like Playing LEGO: Pushing the Modularity of LoRA to Extremes Through Rank-Wise Clustering"
_ICLR.cc/2025/Conference — ICLR 2025 Poster_

### Official Review · Reviewer_YS49 · 2024-11-01

**Soundness:** 2
**Presentation:** 2
**Contribution:** 2
**Rating:** 6
**Confidence:** 2

**Summary:**

This paper introduces the concept of Minimal Semantic Unit (MSU) for disassembling and reassembling multiple LoRAs at a finer granularity. MSUs have permutation invariance and concatenation summation equivalence properties and thus enable flexible combinations for create new LoRAs. Evaluation results show the MSU-based approach achieves a good performance.

**Strengths:**

+ MSU is an interesting concept with permutation invariance and concatenation summation equivalence properties.

**Weaknesses:**

- While MSU is an interesting concept, it is unclear why the finer granularity merging of LoRAs is the right approach for combing multiple LoRAs to enhance LLM capabilities: the clustering and the use of the centroid of each cluster essentially impact the performance of the merged LoRA.
- The evaluation used only 7 LoRAs. It is unclear how the proposed approach may work with more LoRAs.

**Questions:**

How can the finer-grained merging of LoRAs preserve the performance of the merged LoRA, particularly in comparison with the approach of selecting individual LoRAs? How many LoRAs can be supported by the MSU-based approach, and what performance it can achieve with more (e.g., tens of) LoRAs?

---

> ### Author Response · Authors · 2024-11-16
> **(1/3) Rebuttal from Authors of Paper2404 to Reviewer YS49**
>
> Thank you for your insightful and valuable feedback. We will address each of your concerns with a point-by-point response.
>
> > Weakness 1: While MSU is an interesting concept, it is unclear why the **finer granularity merging** of LoRAs is the **right approach for combing multiple LoRAs** to enhance LLM capabilities: the clustering and the use of the centroid of each cluster essentially impact the performance of the merged LoRA.
> >
>
> Thank you for your valuable feedback. In our paper, we identified parameter misalignment and knowledge conflict as the primary factors causing parameter interference during LoRA merging and demonstrated how these factors can significantly degrade merging performance (detailed in Section 2.3 in our paper). Our MSU-based clustering approach effectively addresses both issues in a novel way.
>
> For parameter misalignment, our clustering-based merging ensures that we combine parameters with similar semantic information. This means we only merge parameters that are as close as possible and likely carry similar semantic information, effectively aligning similar parameters before aggregation. Regarding knowledge conflict, our method's flexible rank adjustment (corresponding to the number of clusters) provides sufficient parameter space to accommodate semantic knowledge from different tasks, thereby avoiding conflicts.
>
> Traditional element-wise merging approaches merge parameters at corresponding positions, ignoring the permutation invariance property where different parameters may represent different semantic information. This approach requires proper alignment to avoid parameter interference. Our method effectively addresses this limitation through MSU-level alignment before merging, making it more flexible and efficient.
>
> To validate this, we conducted additional experiments in Appendix F with highly diverse tasks (RTE, CoPA, IMDB, MRPC, BoolQ, CosmosQA, and ARC) where parameter interference is more pronounced. The results show that LoRA-LEGO significantly outperforms other methods, achieving an **11%** improvement over DELLA-Merging, demonstrating effective mitigation of parameter interference through MSU-level alignment and merging:
>
> | Method | RTE | CoPA | IMDB | MRPC | BoolQ | CosmosQA | ARC(Easy) | Average |
> | --- | --- | --- | --- | --- | --- | --- | --- | --- |
> | Weight Average | 47.41 | 51.00 | 5.90 | 56.50 | 60.70 | 20.80 | 12.70 | 36.43 |
> | Ensemble | 52.59 | 3.00 | 33.20 | 35.00 | 73.50 | 0.50 | 19.10 | 30.98 |
> | Ties-Merging | 38.89 | 23.00 | 0.00 | 0.00 | 10.20 | 8.90 | 1.70 | 11.81 |
> | DARE | 47.41 | 53.00 | 6.40 | 57.25 | 60.70 | 21.20 | 12.90 | 36.98 |
> | DELLA-Merging | 55.56 | 63.00 | 58.99 | 68.25 | 72.63 | 18.02 | 38.90 | 53.62 |
> | **LoRA-LEGO** | **56.30** | **69.00** | **65.10** | **60.00** | **80.40** | **35.50** | **50.40** | **59.53** |
>
> These comprehensive results demonstrate our method's superior ability to mitigate parameter interference and effectively combine different LoRAs to enhance LLM capabilities.

---

> ### Author Response · Authors · 2024-11-16
> **(2/3) Rebuttal from Authors of Paper2404 to Reviewer YS49**
>
> > Weakness 2: The evaluation used only 7 LoRAs. It is unclear how the proposed approach may work with more LoRAs.
> >
> > Question 2: How many LoRAs can be supported by the MSU-based approach, and what performance it can achieve with more (e.g., tens of) LoRAs?
> >
>
> Thank you for raising this important concern about scalability. We acknowledge that LoRA merging becomes increasingly challenging with a larger number of LoRAs due to more severe parameter interference. As explained in our response to Weakness 1, our method's ability to effectively mitigate parameter interference makes it particularly robust in large-scale merging scenarios.
>
> To thoroughly evaluate scalability, we conducted comprehensive experiments in **Appendix G**, where we tested merging 20 LoRAs from different tasks. The experimental results reveal several key findings in large-scale, multi-domain LoRA merging scenarios:
>
> 1. Traditional element-wise merging methods show significant performance degradation or complete failure due to severe parameter interference
> 2. The ensemble method failed entirely because of substantial differences in output spaces across multiple LoRA models, leading to irreconcilable conflicts
> 3. DELLA-Merging, while achieving the best performance among baselines through parameter pruning, still struggled to fully address parameter misalignment and knowledge conflicts
> 4. LoRA-LEGO demonstrated consistently superior performance by effectively structuring and integrating both LoRA knowledge and parameters through MSU, successfully resolving parameter interference issues, and achieving a significant **16%** improvement over the best baseline
>
> The detailed results across 20 different tasks are shown in the following table:
>
> | Task | Weight Average | Ensemble | Ties-Merging | DARE | DELLA-Merging | LoRA-LEGO |
> | --- | --- | --- | --- | --- | --- | --- |
> | ARC (Easy) | 1.00 | 0.00 | 3.40 | 1.30 | 37.00 | **50.20** |
> | ARC (Challenge) | 1.30 | 0.00 | 2.50 | 1.30 | 27.90 | **35.20** |
> | CoLA | 0.00 | 0.00 | 0.20 | 0.00 | 54.30 | **54.60** |
> | QQP | 0.00 | 0.00 | 0.20 | 0.00 | **60.20** | 57.30 |
> | PiQA | 1.60 | 0.00 | 7.10 | 1.20 | 45.10 | **45.60** |
> | SST2 | 1.38 | 0.00 | 7.24 | 2.07 | 55.40 | **74.02** |
> | CoPA | 18.00 | 0.00 | 3.00 | 21.00 | 59.00 | **70.00** |
> | IMDB | 0.00 | 0.00 | 1.80 | 0.00 | 51.20 | **64.50** |
> | QNLI | 14.70 | 0.00 | 0.90 | 15.30 | 50.30 | **52.50** |
> | StoryCloze | 40.00 | 0.00 | 5.50 | 40.10 | 63.90 | **64.00** |
> | BoolQ | 12.80 | 0.00 | 5.80 | 13.00 | 69.20 | **81.80** |
> | CosmosQA | 9.30 | 0.00 | 3.90 | 9.70 | 16.40 | **16.50** |
> | MNLI (m) | 5.90 | 0.00 | 0.60 | 5.90 | 39.70 | **59.10** |
> | MNLI (mm) | 3.60 | 0.00 | 0.30 | 3.10 | 40.10 | **56.20** |
> | RTE | 39.63 | 0.00 | 3.70 | 39.63 | 57.78 | **62.22** |
> | STSB | 0.20 | 0.00 | 0.40 | 0.20 | 15.70 | **17.80** |
> | CB | 12.00 | 0.00 | 4.00 | 8.00 | **74.00** | **74.00** |
> | MRPC | 0.25 | 0.00 | 0.75 | 0.25 | 59.00 | **66.75** |
> | SNLI | 0.50 | 0.00 | 0.70 | 0.80 | 35.80 | **55.10** |
> | WNLI | 5.71 | 0.00 | 0.00 | 8.57 | 48.57 | **57.14** |
> | **Average** | 8.39 | 0.00 | 2.60 | 8.57 | 48.03 | **55.73** |
>
> Furthermore, this large-scale evaluation validates our method's ability to handle parameter interference effectively, even when merging a substantial number of LoRAs from diverse domains. We believe these results effectively address the concerns about LoRA-LEGO's scalability and demonstrate its potential for large-scale LoRA integration.

---

> ### Author Response · Authors · 2024-11-16
> **(3/3) Rebuttal from Authors of Paper2404 to Reviewer YS49**
>
> > Question 1: How can the finer-grained merging of LoRAs preserve the performance of the merged LoRA, particularly in comparison with the approach of selecting individual LoRAs?
> >
>
> **Response:**
> Thank you for this insightful question. We can address this from two perspectives: how LoRA-LEGO preserves the capabilities of original LoRAs, and how it compares with the individual LoRA selection approach.
>
> Building upon our previous discussion, the goal of LoRA merging is to preserve all models' capabilities to create a more comprehensive model. However, existing methods often fail to achieve this due to parameter interference, which disrupts the internal structure and capabilities of LoRAs. Our approach addresses this by modularizing LoRAs into finer-grained MSUs, identifying minimal semantic units that can preserve LoRA's internal capabilities while enabling flexible combinations. By aligning the most similar MSUs before merging and implementing flexible rank adjustment, we ensure sufficient parameter space for preserving task-specific knowledge. This approach effectively minimizes parameter interference and preserves the original LoRA capabilities. In some cases, as shown in Table 3 of our paper, the merged model even outperforms individual LoRAs on specific tasks (e.g., RTE), demonstrating how knowledge transfer between similar tasks can enhance overall performance.
>
> Regarding comparison with individual LoRA selection, our paper's Table 4 presents a comprehensive comparison between LoRA-LEGO and the selection approach (selecting top-1 retrieved LoRA). The results show that in **IID** scenarios (where input prompts have corresponding task LoRAs), while selection performs reasonably well, LoRA-LEGO achieves better average performance by leveraging knowledge transfer between similar tasks. In **OOD** scenarios (where input prompts lack corresponding task LoRAs), the Selection method's performance significantly degrades compared to LoRA-LEGO. These results further demonstrate LoRA-LEGO's flexibility and effectiveness in preserving and combining model capabilities across different scenarios, offering a more robust solution than individual LoRA selection.

---

> > ### Comment · Reviewer_YS49 · 2024-11-25
> >
> > Thanks for the response. I raised my socre to 6.

---

> > > ### Author Response · Authors · 2024-11-25
> > >
> > > Thank you for your reply. We're glad to have addressed your concerns and appreciate your helpful suggestions. We’re open to any further discussions and grateful for your time and effort in the review process.

---

### Official Review · Reviewer_Uiyw · 2024-11-01

**Soundness:** 3
**Presentation:** 3
**Contribution:** 3
**Rating:** 6
**Confidence:** 2

**Summary:**

The paper presents an innovative framework called LoRA-LEGO, which enhances the merging of LoRA for large language models. The approach leverages the modularity of LoRAs by introducing Minimal Semantic Units (MSUs) and employs rank-wise clustering to optimize parameter efficiency and performance. The framework aims to overcome limitations in existing methods by reducing parameter interference and maintaining task-specific knowledge.

**Strengths:**

- The concept of treating LoRA parameters as LEGO blocks, specifically through MSUs, is creative and offers a fresh perspective on modularity in model adaptation.
- The paper provides solid theoretical foundations, including permutation invariance and concatenation-summation equivalence, which support the proposed method.
- Extensive experiments demonstrate the framework's effectiveness across multiple benchmarks, outperforming existing methods.

**Weaknesses:**

- The MSU is not related to some semantics that can be understood by humans, it would be better if we could see some analysis.
- Although the proposed method outperforms existing approaches, a more detailed comparison with a wider range of baseline methods could strengthen the claims.
- It does not report the overhead (e.g., in inference and training time, memory) that may introduced by this approach.

**Questions:**

see weakness.

---

> ### Author Response · Authors · 2024-11-16
> **(1/2) Rebuttal from Authors of Paper2404 to Reviewer Uiyw**
>
> Thank you for your careful and insightful feedback. We will address each of your concerns with a point-by-point response.
>
> > Weakness 1: The MSU is not related to some semantics that can be understood by humans, it would be better if we could see some analysis.
> >
> **Response**:
> While directly linking MSUs to specific human-interpretable concepts remains challenging, as it would require manually constructing concept-specific datasets and analyzing their relationships with LoRA parameters, we have conducted alternative analyses to investigate the semantic structure of MSUs.
>
> To explore whether MSUs capture structural semantic information, we conducted a comprehensive analysis using 48 LoRA models, each trained individually on different datasets from Flan-v2 and grouped into 10 distinct task clusters. We visualized the distribution of MSUs using t-SNE projections of the first layer parameters from these LoRAs (detailed visualization provided in **Appendix J**).
>
> Our analysis reveals two key findings:
>
> 1. MSUs within a single LoRA model cluster more closely together, reflecting their shared data distribution
> 2. LoRA models from similar task domains exhibit MSU proximity in the t-SNE space
>
> These patterns suggest that MSUs do capture meaningful task-related information, supporting our approach of considering MSU similarity in the merging process to reduce parameter interference and preserve semantic integrity.
>
>
> > Weakness 2: Although the proposed method outperforms existing approaches, a more detailed comparison with a **wider range of baseline methods** could strengthen the claims.
> >
>
> Thank you for this suggestion. To ensure a comprehensive evaluation, we have strengthened our comparison by including two additional established baselines: **DARE**[1] and **DELLA-Merging**[2]. Our evaluation spans multiple scenarios and model scales to thoroughly validate our method's effectiveness.
>
> First, we present results across multiple tasks using both Llama2-7b and Llama2-13b models (Table 3 in the revised manuscripture). For Llama2-7b, LoRA-LEGO achieves the best average performance (62.21%). This advantage extends to Llama2-13b, where our method maintains superior performance (70.27%) with significant improvements in tasks like SNLI (58.54%) and WNLI (64.29%).
>
> | Method | CoLA | MNLI | MRPC | QNLI | QQP | RTE | SST2 | SNLI | WNLI | Average |
> | --- | --- | --- | --- | --- | --- | --- | --- | --- | --- | --- |
> | **IID Tasks** |  |  |  |  |  |  |  |  |  |  |
> | **w/ Llama2-7b** |  |  |  |  |  |  |  |  |  |  |
> | Task LoRA | 61.63 | 77.46 | 68.00 | 77.25 | 75.83 | 52.22 | 75.74 | N/A | N/A | N/A |
> | Weight Average | 54.42 | 36.09 | **68.00** | 44.41 | 51.72 | 48.15 | 42.99 | 31.64 | 47.14 | 47.17 |
> | Ensemble | **55.67** | 45.89 | 59.25 | 59.84 | 67.38 | 68.89 | 66.44 | 36.73 | 51.43 | 56.84 |
> | Task Arithmetic | 55.48 | 42.15 | 54.25 | 58.94 | 66.43 | 67.78 | 59.54 | 34.08 | 54.29 | 54.77 |
> | Ties-Merging | 48.65 | 48.81 | 55.50 | 61.79 | 66.75 | 62.59 | 70.69 | 48.45 | **61.43** | 58.30 |
> | DARE | 54.62 | 36.16 | 67.75 | 44.41 | 51.83 | 47.78 | 43.45 | 31.64 | 47.14 | 47.20 |
> | DELLA-Merging | 55.19 | 36.88 | 53.25 | 56.04 | 65.69 | 60.37 | 57.70 | 31.02 | 51.43 | 51.95 |
> | LoRA-LEGO | 55.48 | **55.73** | 66.00 | **62.29** | **71.07** | **71.85** | **73.22** | **51.36** | 52.86 | **62.21** |
> | **w/ Llama2-13b** |  |  |  |  |  |  |  |  |  |  |
> | Task LoRA | 69.04 | 88.23 | 89.25 | 82.33 | 86.29 | 80.74 | 76.44 | N/A | N/A | N/A |
> | Weight Average | 45.48 | 46.32 | 67.75 | 46.68 | 47.50 | 62.96 | 46.78 | 42.42 | 42.86 | 49.86 |
> | Ensemble | 62.50 | 64.64 | 74.75 | 71.81 | 81.35 | **79.26** | 75.52 | 54.32 | 60.00 | 69.35 |
> | Task Arithmetic | **63.17** | 64.41 | 74.50 | 71.59 | 80.84 | 78.15 | 75.86 | 54.16 | 58.57 | 69.03 |
> | Ties-Merging | 58.56 | 64.71 | **78.75** | **74.27** | 80.71 | 76.67 | 75.40 | 56.02 | **61.43** | 69.61 |
> | DARE | 45.00 | 46.34 | 67.75 | 46.74 | 47.32 | 63.33 | 46.90 | 42.55 | 44.29 | 50.02 |
> | DELLA-Merging | 62.21 | 62.45 | 71.25 | 69.05 | 76.20 | 78.52 | 75.40 | 49.86 | 58.57 | 67.06 |
> | LoRA-LEGO | 59.42 | **65.40** | 75.50 | 72.29 | **82.51** | 78.52 | **75.98** | **58.54** | **64.29** | **70.27** |
>
> _Continued on next page..._

---

> ### Author Response · Authors · 2024-11-16
> **(2/2) Rebuttal from Authors of Paper2404 to Reviewer Uiyw**
>
> _Continued from the previous page..._
>
> To further validate our method's effectiveness in more challenging scenarios, we conducted additional experiments detailed in **Appendices F&G**:
>
> In **Appendix F**, we evaluated performance across highly diverse tasks (RTE, CoPA, IMDB, MRPC, BoolQ, CosmosQA, and ARC). LoRA-LEGO significantly outperforms other methods, achieving an **11%** improvement over DELLA-Merging, demonstrating effective mitigation of parameter interference through MSU-level alignment and merging.
>
> | Method | RTE | CoPA | IMDB | MRPC | BoolQ | CosmosQA | ARC(Easy) | Average |
> | --- | --- | --- | --- | --- | --- | --- | --- | --- |
> | Weight Average | 47.41 | 51.00 | 5.90 | 56.50 | 60.70 | 20.80 | 12.70 | 36.43 |
> | Ensemble | 52.59 | 3.00 | 33.20 | 35.00 | 73.50 | 0.50 | 19.10 | 30.98 |
> | Ties-Merging | 38.89 | 23.00 | 0.00 | 0.00 | 10.20 | 8.90 | 1.70 | 11.81 |
> | DARE | 47.41 | 53.00 | 6.40 | 57.25 | 60.70 | 21.20 | 12.90 | 36.98 |
> | DELLA-Merging | 55.56 | 63.00 | 58.99 | 68.25 | 72.63 | 18.02 | 38.90 | 53.62 |
> | **LoRA-LEGO** | **56.30** | **69.00** | **65.10** | **60.00** | **80.40** | **35.50** | **50.40** | **59.53** |
>
> In **Appendix G**, we tested scalability by merging 20 LoRAs from different tasks. While traditional methods showed significant degradation or complete failure, and DELLA-Merging struggled with parameter misalignment, LoRA-LEGO maintained robust performance with a **16%** improvement over the best baseline.
>
> | Task | Weight Average | Ensemble | Ties-Merging | DARE | DELLA-Merging | LoRA-LEGO |
> | --- | --- | --- | --- | --- | --- | --- |
> | ARC (Easy) | 1.00 | 0.00 | 3.40 | 1.30 | 37.00 | **50.20** |
> | ARC (Challenge) | 1.30 | 0.00 | 2.50 | 1.30 | 27.90 | **35.20** |
> | CoLA | 0.00 | 0.00 | 0.20 | 0.00 | 54.30 | **54.60** |
> | QQP | 0.00 | 0.00 | 0.20 | 0.00 | **60.20** | 57.30 |
> | PiQA | 1.60 | 0.00 | 7.10 | 1.20 | 45.10 | **45.60** |
> | SST2 | 1.38 | 0.00 | 7.24 | 2.07 | 55.40 | **74.02** |
> | CoPA | 18.00 | 0.00 | 3.00 | 21.00 | 59.00 | **70.00** |
> | IMDB | 0.00 | 0.00 | 1.80 | 0.00 | 51.20 | **64.50** |
> | QNLI | 14.70 | 0.00 | 0.90 | 15.30 | 50.30 | **52.50** |
> | StoryCloze | 40.00 | 0.00 | 5.50 | 40.10 | 63.90 | **64.00** |
> | BoolQ | 12.80 | 0.00 | 5.80 | 13.00 | 69.20 | **81.80** |
> | CosmosQA | 9.30 | 0.00 | 3.90 | 9.70 | 16.40 | **16.50** |
> | MNLI (m) | 5.90 | 0.00 | 0.60 | 5.90 | 39.70 | **59.10** |
> | MNLI (mm) | 3.60 | 0.00 | 0.30 | 3.10 | 40.10 | **56.20** |
> | RTE | 39.63 | 0.00 | 3.70 | 39.63 | 57.78 | **62.22** |
> | STSB | 0.20 | 0.00 | 0.40 | 0.20 | 15.70 | **17.80** |
> | CB | 12.00 | 0.00 | 4.00 | 8.00 | **74.00** | **74.00** |
> | MRPC | 0.25 | 0.00 | 0.75 | 0.25 | 59.00 | **66.75** |
> | SNLI | 0.50 | 0.00 | 0.70 | 0.80 | 35.80 | **55.10** |
> | WNLI | 5.71 | 0.00 | 0.00 | 8.57 | 48.57 | **57.14** |
> | **Average** | 8.39 | 0.00 | 2.60 | 8.57 | 48.03 | **55.73** |
>
> These comprehensive evaluations across different settings (model scales, task diversity, and large-scale merging) demonstrate our method's consistent superiority in handling parameter interference and maintaining performance in challenging scenarios. We believe these extensive comparisons thoroughly validate the effectiveness of our approach.
>
> [1] Yu, Le, et al. "Language models are super mario: Absorbing abilities from homologous models as a free lunch." *Forty-first International Conference on Machine Learning*. 2024.
>
> [2] Deep, Pala Tej, Rishabh Bhardwaj, and Soujanya Poria. "DELLA-Merging: Reducing Interference in Model Merging through Magnitude-Based Sampling." *arXiv preprint arXiv:2406.11617* (2024).
>
> > Weakness 3:  It does not report the overhead (e.g., in inference and training time, memory) that may introduced by this approach.
> >
>
> We have conducted detailed overhead analysis using an A100 GPU with a batch size of 16. For fair comparison, we measured inference time based on first-token generation, as output lengths may vary across models. Importantly, LoRA-LEGO is training-free, incurring no additional training overhead.
>
> Here are the detailed results:
>
> | **Method** | **Average Inference Time (seconds)** | **Peak GPU Memory Consumption (MB)** | **Average Accuracy (%)** |
> | --- | --- | --- | --- |
> | **LoRA-LEGO** | 0.1943 | 13230.86 | 62.21 |
> | DARE | 0.1982 | 13222.86 | 47.20 |
> | DELLA-Merging | 0.1831 | 13344.66 | 51.95 |
> | Ensemble | 0.2320 | 13392.66 | 56.84 |
> | Weight Average | 0.1931 | 13344.66 | 47.17 |
> | Ties-Merging | 0.1876 | 13344.99 | 58.30 |
>
> As shown, LoRA-LEGO achieves competitive inference performance with inference time comparable to other methods, similar memory consumption, and significantly higher accuracy (62.21%). These results demonstrate that LoRA-LEGO delivers superior performance without introducing significant computational overhead, making it both effective and efficient in practical applications.

---

> > ### Comment · Reviewer_Uiyw · 2024-11-25
> > **Thank you for the rebuttal**
> >
> > I have read and reviewed the update and would like to maintain my score. The visualization of MSU still does not convince me, but the additional baselines seem good. However, there are also some concerns about the sensitivity of the hyper-parameters.

---

> > > ### Author Response · Authors · 2024-11-25
> > >
> > > Thank you for your reply and we appreciate you acknowledging the additional experiments and baselines we included in our revised manuscript.
> > >
> > > Regarding your concern about whether **MSUs can be directly linked to human-understandable concepts**, we'd like to provide some further clarification. Based on the properties of MSU **permutation invariance** and **concatenation-summation equivalence**, we conclude that all MSUs are independent of each other. Our visualizations confirm discernible patterns: MSUs within the same task cluster more closely together, and those from similar tasks are also closer in the representation space.
> > >
> > > While further decoupling and associating different MSUs with specific concepts is beyond the scope of our current work, we agree that this is an intriguing direction for future research. For example, one could explore constraining different MSUs during LoRA training to focus on distinct concepts and knowledge areas, potentially enhancing LoRA's overall capabilities.
> > >
> > > As for the **sensitivity to hyper-parameters**, as we mentioned in our response to Reviewer **`9LLA`**'s Weakness 2, our method is highly stable and robust concerning hyper-parameter selection. The only hyper-parameter in our method is the cluster number *k*, which corresponds to the rank of the merged LoRA.
> > >
> > > This robustness stems from our discovery of a general relationship between the LoRA output scale and the merged rank, as illustrated in Figure 7 of our paper. After implementing our dual-reweighting strategy (shown by the red line in Figure 7), our method maintains stable performance across different cluster numbers *k*. In contrast, without dual-reweighting, performance varies significantly due to output scale variations.
> > >
> > > This inherent stability allows us to use fixed cluster numbers across all experiments while maintaining consistent performance. The dual-reweighting strategy effectively addresses the scale variation issue that typically makes merging methods sensitive to hyper-parameter choices. We have expanded the "**Ablation of Scaling Strategies**" section to include a detailed discussion of hyper-parameter robustness and the role of dual-reweighting in achieving this stability.
> > >
> > > Thank you again for your insightful comments and for helping us improve our work.

---

> > > > ### Author Response · Authors · 2024-12-02
> > > > **Follow-up on Recent Response**
> > > >
> > > > Dear Reviewer **`Uiyw`,**
> > > >
> > > > As the discussion phase is coming to an end, we would be very grateful if you could check whether our latest response has successfully addressed your concerns. If you have any remaining questions or would like to discuss any points further, we would deeply appreciate your additional feedback.
> > > >
> > > > We deeply value your feedback and want to ensure all issues are properly resolved before the discussion concludes.
> > > >
> > > > Thank you very much for your time and valuable input throughout this process.
> > > >
> > > > Best regards,
> > > >
> > > > The Authors

---

### Official Review · Reviewer_9LLA · 2024-11-02

**Soundness:** 4
**Presentation:** 3
**Contribution:** 3
**Rating:** 6
**Confidence:** 3

**Summary:**

This paper proposed the LoRA-LEGO framework for combining Low-Rank Adaptation (LoRA) modules in large language models (LLMs). LoRA-LEGO introduces Minimal Semantic Units (MSUs), independent components within LoRA, enabling flexible merging through clustering and rank adjustment. This modular approach allows task-specific LoRAs to be disassembled and reassembled, tackling parameter misalignment and knowledge conflicts seen in traditional merging methods. The dual reweighting strategy optimizes the scale of the combined LoRA, resulting in a more efficient and flexible system that performs well across both in-domain and out-of-domain tasks.

**Strengths:**

* Modularity and Flexibility: The MSU-based modular approach enhances flexibility in merging LoRAs, allowing for custom rank adjustments.

* Efficient Scaling: Dual reweighting improves output scale management, maintaining model performance across tasks.

**Weaknesses:**

* Performance may degrade when merging LoRAs designed for highly divergent tasks.

* Merging effectiveness can be highly sensitive to hyperparameter settings.

**Questions:**

1. How does LoRA-LEGO address performance degradation when merging LoRAs for tasks with significant divergence?

2. Why does performance improve for MRPC and RTE tasks in Table 1 following permutation merging?

3. How can the optimal number of clusters $k$ in equation (2) be determined?

4. Why does the weight averaging method in Table 3 achieve identical performance to the task-specific LoRA (upper bound) for the MRPC task?

---

> ### Author Response · Authors · 2024-11-16
> **(1/3) Rebuttal from Authors of Paper2404 to Reviewer 9LLA**
>
> Thank you for your thoughtful questions and constructive feedback. We will address each of your concerns with a point-by-point response.
>
> > Weakness 1: Performance may degrade when merging LoRAs designed for highly divergent tasks.
> >
> > Question 1: How does LoRA-LEGO address performance degradation when merging LoRAs for tasks with significant divergence?
> >
>
> **Response:**
> We acknowledge that highly divergent tasks present a significant challenge for model merging methods due to increased parameter interference. LoRA-LEGO performs merging by clustering MSUs, which first ensures that all aggregated parameters are sufficiently aligned by grouping similar ones. Through this approach, parameter alignment is achieved. Additionally, the flexibility of the clustering algorithm allows us to freely choose the rank of the merged LoRA, ensuring that the final LoRA has enough parameter space to avoid knowledge conflict by adjusting the rank. These features provide significant advantages over traditional element-wise merging methods.
>
> To validate our method's effectiveness with diverse tasks, we first conducted experiments in **Appendix F**  using LoRAs trained on various tasks:
>
> - RTE (natural language inference)
> - CoPA (commonsense reasoning)
> - IMDB (sentiment analysis)
> - MRPC (paraphrase detection)
> - BoolQ (reading comprehension)
> - CosmosQA (reading comprehension with commonsense)
> - ARC (closed-book question answering)
>
> These tasks represent significantly different domains and objectives. The results are presented in Table 1:
>
> **Table 1: Performance on Diverse Tasks**
>
> | Method | RTE | CoPA | IMDB | MRPC | BoolQ | CosmosQA | ARC(Easy) | Average |
> | --- | --- | --- | --- | --- | --- | --- | --- | --- |
> | Weight Average | 47.41 | 51.00 | 5.90 | 56.50 | 60.70 | 20.80 | 12.70 | 36.43 |
> | Ensemble | 52.59 | 3.00 | 33.20 | 35.00 | 73.50 | 0.50 | 19.10 | 30.98 |
> | Ties-Merging | 38.89 | 23.00 | 0.00 | 0.00 | 10.20 | 8.90 | 1.70 | 11.81 |
> | DARE | 47.41 | 53.00 | 6.40 | 57.25 | 60.70 | 21.20 | 12.90 | 36.98 |
> | DELLA-Merging | 55.56 | 63.00 | 58.99 | 68.25 | 72.63 | 18.02 | 38.90 | 53.62 |
> | **LoRA-LEGO** | **56.30** | **69.00** | **65.10** | **60.00** | **80.40** | **35.50** | **50.40** | **59.53** |
>
> To further evaluate scalability, we conducted more challenging experiments merging 20 LoRAs from diverse tasks in Appendix G. The results are shown in Table 2:
>
> **Table 2: Large-scale Integration of 20 LoRAs**
>
> | Task | Weight Average | Ensemble | Ties-Merging | DARE | DELLA-Merging | LoRA-LEGO |
> | --- | --- | --- | --- | --- | --- | --- |
> | ARC (Easy) | 1.00 | 0.00 | 3.40 | 1.30 | 37.00 | **50.20** |
> | ARC (Challenge) | 1.30 | 0.00 | 2.50 | 1.30 | 27.90 | **35.20** |
> | CoLA | 0.00 | 0.00 | 0.20 | 0.00 | 54.30 | **54.60** |
> | QQP | 0.00 | 0.00 | 0.20 | 0.00 | **60.20** | 57.30 |
> | PiQA | 1.60 | 0.00 | 7.10 | 1.20 | 45.10 | **45.60** |
> | SST2 | 1.38 | 0.00 | 7.24 | 2.07 | 55.40 | **74.02** |
> | CoPA | 18.00 | 0.00 | 3.00 | 21.00 | 59.00 | **70.00** |
> | IMDB | 0.00 | 0.00 | 1.80 | 0.00 | 51.20 | **64.50** |
> | QNLI | 14.70 | 0.00 | 0.90 | 15.30 | 50.30 | **52.50** |
> | StoryCloze | 40.00 | 0.00 | 5.50 | 40.10 | 63.90 | **64.00** |
> | BoolQ | 12.80 | 0.00 | 5.80 | 13.00 | 69.20 | **81.80** |
> | CosmosQA | 9.30 | 0.00 | 3.90 | 9.70 | 16.40 | **16.50** |
> | MNLI (m) | 5.90 | 0.00 | 0.60 | 5.90 | 39.70 | **59.10** |
> | MNLI (mm) | 3.60 | 0.00 | 0.30 | 3.10 | 40.10 | **56.20** |
> | RTE | 39.63 | 0.00 | 3.70 | 39.63 | 57.78 | **62.22** |
> | STSB | 0.20 | 0.00 | 0.40 | 0.20 | 15.70 | **17.80** |
> | CB | 12.00 | 0.00 | 4.00 | 8.00 | **74.00** | **74.00** |
> | MRPC | 0.25 | 0.00 | 0.75 | 0.25 | 59.00 | **66.75** |
> | SNLI | 0.50 | 0.00 | 0.70 | 0.80 | 35.80 | **55.10** |
> | WNLI | 5.71 | 0.00 | 0.00 | 8.57 | 48.57 | **57.14** |
> | **Average** | 8.39 | 0.00 | 2.60 | 8.57 | 48.03 | **55.73** |
>
> These comprehensive experiments reveal several key findings:
>
> 1. In the initial diverse task setting, traditional methods suffer from severe performance degradation, while LoRA-LEGO achieves an **11%** improvement over the best baseline.
> 2. In the 20-LoRA scenario, the challenges become more pronounced:
>     - Element-wise merging methods show significant performance drops or complete failure
>     - The ensemble method fails entirely due to conflicts in output spaces
>     - DELLA-Merging, while performing better than other baselines by using magnitude pruning to reduce the impact of large task numbers and task diversity, still fails to fully address the fundamental challenges arising from parameter misalignment and knowledge conflict.
>     - LoRA-LEGO maintains robust performance, achieving a **16%** improvement over the best baseline
>
> These results demonstrate that LoRA-LEGO effectively addresses the challenge of merging LoRAs from highly divergent tasks, maintaining strong performance even in scenarios with significant task diversity and scale.

---

> ### Author Response · Authors · 2024-11-16
> **(2/3) Rebuttal from Authors of Paper2404 to Reviewer 9LLA**
>
> > Weakness 2: Merging effectiveness can be highly sensitive to hyperparameter settings.
> >
> >Question 3: How can the optimal number of clusters k in equation (2) be determined?
> >
>
> **Response:**
> Thank you for raising this important concern about hyperparameter sensitivity. We would like to emphasize that LoRA-LEGO demonstrates remarkable robustness to hyperparameter settings. The only hyperparameter in our method is the cluster number $k$, which corresponds to the rank of the merged LoRA.
>
> This robustness stems from our discovery of a general relationship between the LoRA output scale and merged rank, as illustrated in Figure 7 of our paper. After implementing our dual-reweighting strategy (shown by the red line in Figure 7), our method maintains stable performance across different cluster numbers $k$. In contrast, without dual-reweighting, performance varies significantly due to output scale variations. This stability can be attributed to two key factors:
>
> 1. Our dual-reweighting strategy effectively normalizes the output scale of merged LoRAs, making the performance more consistent across different cluster numbers
> 2. The MSU-based clustering approach naturally groups similar parameters together, making the method less sensitive to the exact choice of $k$
>
> This inherent stability allows us to use fixed cluster numbers across all experiments while maintaining consistent performance. The dual-reweighting strategy effectively addresses the scale variation issue that typically makes merging methods sensitive to hyperparameter choices. We have expanded the "**Ablation of Scaling Strategies**" section to include a detailed discussion of hyperparameter robustness and the role of dual-reweighting in achieving this stability.
>
> > Question 2: Why does performance improve for MRPC and RTE tasks in Table 1 following permutation merging?
> >
>
> **Response:**
> Thank you for this insightful question. The performance improvement in MRPC and RTE tasks can be attributed to the unique characteristics of these tasks and their internal MSU structures. We have conducted a detailed analysis to understand this phenomenon.
>
> In **Appendix I**, we plotted the average Euclidean distances between MSUs in each layer across all LoRAs. The analysis reveals that MRPC and RTE exhibit notably smaller internal MSU distances compared to other tasks. This smaller distance indicates:
>
> 1. These tasks are relatively simple with smaller datasets
> 2. Their internal MSUs show higher similarity
> 3. Their parameters potentially contain redundancy and may be prone to overfitting
>
> When applying permutation merging $(A+PA)/2, (B+BP^T)/2$, it effectively acts as a form of **parameter smoothing** or **regularization**. For tasks like MRPC and RTE, where MSUs are more closely connected (as evidenced by smaller distances), this regularization actually helps reduce overfitting, leading to performance improvements. However, for tasks where MSUs are more distant from each other, such permutation merging can cause significant performance degradation.
>
> This observation further validates the importance of considering MSU connectivity in LoRA merging strategies. Our clustering-based approach in LoRA-LEGO specifically accounts for these connectivity patterns, merging more closely aligned MSUs to effectively mitigate parameter interference while preserving task-specific knowledge. This explains why LoRA-LEGO achieves consistent performance improvements across diverse tasks, regardless of their internal MSU structure.

---

> ### Author Response · Authors · 2024-11-16
> **(3/3) Rebuttal from Authors of Paper2404 to Reviewer 9LLA**
>
> > Question 4: Why does the weight averaging method in Table 3 achieve identical performance to the task-specific LoRA (upper bound) for the MRPC task?
> >
>
> **Response:**
> Thank you for this astute observation. This interesting phenomenon can be understood as an extension of our discussion from Question 2 regarding the characteristics of the MRPC task. To investigate this further, we conducted an additional experiment to verify our hypothesis about knowledge transfer between tasks.
>
> We performed two sets of merging experiments:
>
> 1. Original weight averaging including all LoRAs
> 2. Weight averaging excluding the MRPC LoRA
>
> The results are shown below:
>
> | Task LoRA | Weight Average | Weight Average w/o MRPC LoRA |
> | --- | --- | --- |
> | 68.00 | 68.00 | 67.25 |
>
> These results reveal several interesting insights:
>
> - Even without including the MRPC LoRA, the merged model achieves nearly identical performance to the task-specific MRPC LoRA
> - This suggests that parameters from other tasks can effectively transfer to the MRPC task
> - The similarity in performance indicates that knowledge from related tasks (e.g., QQP, which also involves sentence pair similarity classification) can complement and potentially enhance MRPC task performance
>
> This phenomenon demonstrates the potential for positive knowledge transfer between similar tasks through model merging, suggesting that well-designed merging strategies can actually improve generalization across related tasks. It also aligns with our earlier findings about MRPC's parameter structure and its amenability to parameter sharing and regularization through merging.

---

> > ### Comment · Reviewer_9LLA · 2024-11-25
> >
> > Thank you for your thoughtful response. While your experimental results address my questions to some extent, I remain interested in the innovative points raised by other reviewers. For now, I will maintain my score and look forward to further discussions between you and the other reviewers on this matter.

---

> > > ### Author Response · Authors · 2024-11-25
> > >
> > > Thank you for your reply. We hope our replies have addressed your concerns. If you have any further suggestions or questions, please feel free to let us know. Once again, we sincerely appreciate your time and effort in reviewing our work.

---

> > > ### Author Response · Authors · 2024-12-02
> > > **Follow-up on Recent Discussions and Responses**
> > >
> > > Dear Reviewer **`9LLA`,**
> > >
> > > Thank you for your previous feedback. As the discussion phase is coming to an end, we wanted to inform you that we have provided comprehensive responses to all concerns and suggestions raised, including the insightful points brought up by other reviewers that you expressed interest in. We would be very grateful if you could review these responses.
> > >
> > > We highly value your opinion and sincerely hope to address any remaining concerns before the discussion phase concludes. If you have any additional thoughts or feedback after reviewing our responses, we would deeply appreciate hearing from you.
> > >
> > > Thank you again for your valuable feedback throughout this process.
> > >
> > > Best regards,
> > >
> > > The Authors

---

### Official Review · Reviewer_1z1o · 2024-11-04

**Soundness:** 3
**Presentation:** 3
**Contribution:** 3
**Rating:** 6
**Confidence:** 4

**Summary:**

The paper introduces the LoRA-LEGO framework, which combines multiple LoRAs for multi-task and mixed-task scenarios. It uses clustering to group MSUs from different LoRAs into k clusters before constructing a merged LoRA based on the cluster centroids. Parameter reweighting is then performed to scale the centroid with respect to the average cluster norm. Experiments are conducted against 4 baselines (weighted average, ensemble, task arithmetic and ties merging) for multi-task and mixed-task scenarios, showing improved average performance.

**Strengths:**

After reading the authors' rebuttals, my concerns have been adequately addressed.

**Weaknesses:**

1. The paper is generally well written with good illustrations. However, I found the approach not particularly novel. It relies on a straightforward k-means clustering method for grouping and subsequently aggregating cluster parameters.

2. The approach is not particularly strong, and this simplicity may limit the framework’s performance which is reflected in the empirical results. While the average performance across tasks improves, it does not apply to all tasks. For example, performance remains poor for WNLI, an out of domain task for multi-task learning.

3. The baselines involving LoRA composition methods are basic and are not established methods in existing literature. It lacks citations that would align them with related work. This weakens the comparative analysis and raises questions about the validity of the selected baselines.

**Questions:**

See above

---

> ### Author Response · Authors · 2024-11-16
> **(1/3) Rebuttal from Authors of Paper2404 to Reviewer 1z1o**
>
> We sincerely thank you for your valuable feedback, and we are grateful for the time and effort you have invested in reviewing our work. Below, we provide a point-by-point response to address each of your concerns:
>
> > Weakness 1: "The paper is generally well written with good illustrations. However, I found the approach not particularly novel. It relies on a straightforward k-means clustering method for grouping and subsequently aggregating cluster parameters."
> >
>
> **Response**:
> We appreciate your feedback and would like to clarify the main contributions and novelty of our approach.
>
> The core contribution of our work lies in the **further modularization of LoRA**, enabled by the novel concept of MSUs and the proposed **MSU clustering** framework for LoRA merging. Prior work has not recognized that LoRA merging can be achieved through finer-grained parameter clustering, and such an approach offers greater flexibility and alleviates **parameter interference**. Moreover, our method can be used for merging heterogeneous LoRAs and applied to a singular LoRA for parameter pruning, which demonstrates the flexibility and universality of our approach across various problems.
>
> Regarding the use of k-means clustering, we respectfully disagree with the notion that its simplicity detracts from the novelty of our approach. **We believe that the true value of a contribution often stems from new insights and innovative perspectives, rather than the complexity of the techniques employed.** Our choice of **simple, yet effective methods** was deliberate, as simplicity enhances the clarity, interpretability, and accessibility of our work, making it more adaptable for the broader community.
>
> We also believe that our method introduces an innovative viewpoint on LoRAs, inspiring potentially impactful research directions in modular model adaptation. For example:
>
> - Continuously adding MSUs to LoRA can expand the model's capabilities for Lifelong Learning
> - Sharing and integrating different LoRA MSUs can enable more efficient LLM Federated Learning
> - Model pruning can be achieved through the integration of internal LoRA MSUs
> - Dynamic adjustment of LoRA MSUs can lead to more efficient LoRA training
>
> Through this unique perspective and our novel approach to LoRA merging via MSU clustering, we believe our method offers significant novelty and inspiration to the field. We hope that this clarification highlights the novelty and potential utility of our approach and addresses your concern regarding our contribution.

---

> ### Author Response · Authors · 2024-11-16
> **(2/3) Rebuttal from Authors of Paper2404 to Reviewer 1z1o**
>
> > Weakness 2: "The approach is not particularly strong, and this simplicity may limit the framework's performance which is reflected in the empirical results. While the average performance across tasks improves, it does not apply to all tasks. For example, performance remains poor for WNLI, an out-of-domain task for multi-task learning."
> >
>
> **Response**:
> Thank you for raising these concerns about our method's performance. We would like to provide additional evidence and clarification.
>
> Our approach demonstrates substantial and statistically significant improvements over baseline methods across several benchmarks. Specifically, on the Llama2-7b model, our method outperforms the best baseline by **6.7%**, and on Llama2-13b, it achieves a **1%** improvement.
>
> Furthermore, we conducted additional experiments outlined in **Appendix F** and **Appendix G** to validate our method in more **diverse** and **scaled** task settings:
>
> 1. **Performance on Diverse Tasks** (Appendix F):
> We evaluated our method's performance when integrating LoRA models from highly diverse tasks, including:
> - RTE (natural language inference)
> - CoPA (commonsense reasoning)
> - IMDB reviews (sentiment analysis)
> - MRPC (paraphrase detection)
> - BoolQ (reading comprehension)
> - CosmosQA (reading comprehension with commonsense)
> - ARC (closed-book question answering)
>
> In this challenging scenario with increased parameter interference due to task diversity, LoRA-LEGO significantly outperforms other methods, achieving an **11%** improvement over the best baseline (DELLA-Merging). This demonstrates our method's effectiveness in mitigating parameter interference through MSU-level alignment and merging.
>
> | Method | RTE | CoPA | IMDB | MRPC | BoolQ | CosmosQA | ARC(Easy) | Average |
> | --- | --- | --- | --- | --- | --- | --- | --- | --- |
> | Weight Average | 47.41 | 51.00 | 5.90 | 56.50 | 60.70 | 20.80 | 12.70 | 36.43 |
> | Ensemble | 52.59 | 3.00 | 33.20 | 35.00 | 73.50 | 0.50 | 19.10 | 30.98 |
> | Ties-Merging | 38.89 | 23.00 | 0.00 | 0.00 | 10.20 | 8.90 | 1.70 | 11.81 |
> | DARE | 47.41 | 53.00 | 6.40 | 57.25 | 60.70 | 21.20 | 12.90 | 36.98 |
> | DELLA-Merging | 55.56 | 63.00 | 58.99 | 68.25 | 72.63 | 18.02 | 38.90 | 53.62 |
> | **LoRA-LEGO** | **56.30** | **69.00** | **65.10** | **60.00** | **80.40** | **35.50** | **50.40** | **59.53** |
> 1. **Scalability Analysis** (Appendix G):
> We further evaluated the scalability of all methods by merging 20 LoRAs from different tasks. The results show that:
> - Traditional element-wise merging methods experience significant performance degradation
> - The ensemble method failed completely due to conflicts in output spaces
> - DELLA-Merging, while performing best among baselines, still struggled with parameter misalignment
> - LoRA-LEGO maintained superior performance, achieving a **16%** improvement over the best baseline
>
> | Task | Weight Average | Ensemble | Ties-Merging | DARE | DELLA-Merging | LoRA-LEGO |
> | --- | --- | --- | --- | --- | --- | --- |
> | ARC (Easy) | 1.00 | 0.00 | 3.40 | 1.30 | 37.00 | **50.20** |
> | ARC (Challenge) | 1.30 | 0.00 | 2.50 | 1.30 | 27.90 | **35.20** |
> | CoLA | 0.00 | 0.00 | 0.20 | 0.00 | 54.30 | **54.60** |
> | QQP | 0.00 | 0.00 | 0.20 | 0.00 | **60.20** | 57.30 |
> | PiQA | 1.60 | 0.00 | 7.10 | 1.20 | 45.10 | **45.60** |
> | SST2 | 1.38 | 0.00 | 7.24 | 2.07 | 55.40 | **74.02** |
> | CoPA | 18.00 | 0.00 | 3.00 | 21.00 | 59.00 | **70.00** |
> | IMDB | 0.00 | 0.00 | 1.80 | 0.00 | 51.20 | **64.50** |
> | QNLI | 14.70 | 0.00 | 0.90 | 15.30 | 50.30 | **52.50** |
> | StoryCloze | 40.00 | 0.00 | 5.50 | 40.10 | 63.90 | **64.00** |
> | BoolQ | 12.80 | 0.00 | 5.80 | 13.00 | 69.20 | **81.80** |
> | CosmosQA | 9.30 | 0.00 | 3.90 | 9.70 | 16.40 | **16.50** |
> | MNLI (m) | 5.90 | 0.00 | 0.60 | 5.90 | 39.70 | **59.10** |
> | MNLI (mm) | 3.60 | 0.00 | 0.30 | 3.10 | 40.10 | **56.20** |
> | RTE | 39.63 | 0.00 | 3.70 | 39.63 | 57.78 | **62.22** |
> | STSB | 0.20 | 0.00 | 0.40 | 0.20 | 15.70 | **17.80** |
> | CB | 12.00 | 0.00 | 4.00 | 8.00 | **74.00** | **74.00** |
> | MRPC | 0.25 | 0.00 | 0.75 | 0.25 | 59.00 | **66.75** |
> | SNLI | 0.50 | 0.00 | 0.70 | 0.80 | 35.80 | **55.10** |
> | WNLI | 5.71 | 0.00 | 0.00 | 8.57 | 48.57 | **57.14** |
> | **Average** | 8.39 | 0.00 | 2.60 | 8.57 | 48.03 | **55.73** |
>
> These comprehensive evaluations demonstrate that our method is particularly effective in scenarios with high task diversity, large-scale parameter merging, and significant parameter interference. The experimental results in Appendices F and G provide strong evidence of our method's robustness and effectiveness, particularly in challenging scenarios where parameter interference is more pronounced. We believe these additional experiments address your concerns about the method's performance limitations.

---

> ### Author Response · Authors · 2024-11-16
> **(3/3) Rebuttal from Authors of Paper2404 to Reviewer 1z1o**
>
> > Weakness 3: "The baselines involving LoRA composition methods are basic and are not established methods in the existing literature. It lacks citations that would align them with related work. This weakens the comparative analysis and raises questions about the validity of the selected baselines."
> >
>
> **Response**:
>
> Thank you for raising this important concern about our baseline selection and related work discussion. We would like to clarify our baseline selection criteria and present additional comparisons.
>
> Given that our method is training-free, we focused on comparing it with other training-free model merging methods to ensure fair comparison. We have strengthened our evaluation by including two additional methods: **DARE[1] and DELLA-Merging[2]**. As demonstrated in the following experimental results, LoRA-LEGO significantly outperforms these strong baselines, further validating our method's effectiveness.
>
> | Method | CoLA | MNLI | MRPC | QNLI | QQP | RTE | SST2 | SNLI | WNLI | Average |
> | --- | --- | --- | --- | --- | --- | --- | --- | --- | --- | --- |
> | **w/ Llama2-7b** |  |  |  |  |  |  |  |  |  |  |
> | Task LoRA | 61.63 | 77.46 | 68.00 | 77.25 | 75.83 | 52.22 | 75.74 | N/A | N/A | N/A |
> | Weight Average | 54.42 | 36.09 | **68.00** | 44.41 | 51.72 | 48.15 | 42.99 | 31.64 | 47.14 | 47.17 |
> | Ensemble | **55.67** | 45.89 | 59.25 | 59.84 | 67.38 | 68.89 | 66.44 | 36.73 | 51.43 | 56.84 |
> | Task Arithmetic | 55.48 | 42.15 | 54.25 | 58.94 | 66.43 | 67.78 | 59.54 | 34.08 | 54.29 | 54.77 |
> | Ties-Merging | 48.65 | 48.81 | 55.50 | 61.79 | 66.75 | 62.59 | 70.69 | 48.45 | **61.43** | 58.30 |
> | DARE | 54.62 | 36.16 | 67.75 | 44.41 | 51.83 | 47.78 | 43.45 | 31.64 | 47.14 | 47.20 |
> | DELLA-Merging | 55.19 | 36.88 | 53.25 | 56.04 | 65.69 | 60.37 | 57.70 | 31.02 | 51.43 | 51.95 |
> | LoRA-LEGO | 55.48 | **55.73** | 66.00 | **62.29** | **71.07** | **71.85** | **73.22** | **51.36** | 52.86 | **62.21** |
> | **w/ Llama2-13b** |  |  |  |  |  |  |  |  |  |  |
> | Task LoRA | 69.04 | 88.23 | 89.25 | 82.33 | 86.29 | 80.74 | 76.44 | N/A | N/A | N/A |
> | Weight Average | 45.48 | 46.32 | 67.75 | 46.68 | 47.50 | 62.96 | 46.78 | 42.42 | 42.86 | 49.86 |
> | Ensemble | 62.50 | 64.64 | 74.75 | 71.81 | 81.35 | **79.26** | 75.52 | 54.32 | 60.00 | 69.35 |
> | Task Arithmetic | **63.17** | 64.41 | 74.50 | 71.59 | 80.84 | 78.15 | 75.86 | 54.16 | 58.57 | 69.03 |
> | Ties-Merging | 58.56 | 64.71 | **78.75** | **74.27** | 80.71 | 76.67 | 75.40 | 56.02 | **61.43** | 69.61 |
> | DARE | 45.00 | 46.34 | 67.75 | 46.74 | 47.32 | 63.33 | 46.90 | 42.55 | 44.29 | 50.02 |
> | DELLA-Merging | 62.21 | 62.45 | 71.25 | 69.05 | 76.20 | 78.52 | 75.40 | 49.86 | 58.57 | 67.06 |
> | LoRA-LEGO | 59.42 | **65.40** | 75.50 | 72.29 | **82.51** | 78.52 | **75.98** | **58.54** | **64.29** | **70.27** |
>
> We have also expanded our discussion of related work in **Appendix H**, providing a more detailed discussion of current model merging methods. We welcome suggestions for additional baseline methods or related work comparisons and are prepared to conduct supplementary experiments during the rebuttal period to address any specific concerns.
>
> [1] Yu, Le, et al. "Language models are super mario: Absorbing abilities from homologous models as a free lunch." *Forty-first International Conference on Machine Learning*. 2024.
>
> [2] Deep, Pala Tej, Rishabh Bhardwaj, and Soujanya Poria. "DELLA-Merging: Reducing Interference in Model Merging through Magnitude-Based Sampling." *arXiv preprint arXiv:2406.11617* (2024).

---

> ### Author Response · Authors · 2024-11-27
> **Any New Comments Would be Greatly Appreciated**
>
> Dear Reviewer `1z1o`,
>
> We sincerely appreciate the valuable suggestions and comments you provided for our paper. We have carefully considered each point and have addressed them in detail in our rebuttal.
>
> As the Author-Review Discussion period draws to a close, we want to ensure that all your concerns have been adequately addressed. If there are any questions or unresolved issues, we are eager to provide further clarification or make necessary revisions.
>
> Best regards,
>
> The Authors

---

> > ### Comment · Reviewer_1z1o · 2024-12-03
> > **Thanks**
> >
> > Thanks for the clarifications. I concerns have been addressed. I will raise my score to 6.

---

> > > ### Author Response · Authors · 2024-12-03
> > >
> > > Thank you for your response. We are delighted to hear that your concerns have been fully addressed and we appreciate your time and effort in reviewing our work.

---

### Author Response · Authors · 2024-11-16
**General Response to the Reviewers**

We sincerely appreciate the reviewers for their thoughtful and constructive feedback. We are encouraged by the positive recognition of our contribution, which can be summarized as follows:

1. MSU is highlighted as an innovative and meaningful concept for modular model adaptation (Reviewers `YS49`, `Uiyw`)
2. Our work establishes solid theoretical foundations through properties like permutation invariance and concatenation-summation equivalence (Reviewers `YS49`, `Uiyw`)
3. The LEGO-like approach provides enhanced flexibility and modularity in LoRA merging (Reviewers `Uiyw`, `9LLA`)
4. Our method demonstrates superior empirical performance across multiple benchmarks with efficient scaling (Reviewers `Uiyw`, `9LLA`)

In our revision, we have carefully addressed each of the concerns raised, and below are the main points of the revised manuscript:

1. We have included **two additional baselines**: DARE [1] and DELLA-Merging [2], to provide more comprehensive comparisons with recent model merging methods. Detailed comparisons and analyses are presented in **Table 3 and Appendix H** of the revision. (Reviewers `1z1o`, `Uiyw`)
2. We have conducted experiments with more **diverse tasks**: In **Appendix F**, we evaluate LoRA merging performance using 7 LoRAs from different domains, where our method significantly outperforms baselines with an 11% improvement over the best baseline. (Reviewers `1z1o`, `9LLA`,  `Uiyw`, `YS49`)
3. We have assessed the **scalability** of different methods: In **Appendix G**, we evaluate the merging capabilities of different methods using 20 LoRAs from various tasks and domains. Our method maintains superior performance with a 16% improvement over the best baseline in this challenging scenario.(Reviewers `1z1o`, `9LLA`,  `Uiyw`, `YS49`)
4. We have investigated MSU similarity within LoRAs: In **Appendix I**, we analyze the similarity of MSUs within different task LoRAs, revealing that simpler tasks tend to have more similar internal MSUs, making them more receptive to positive ability transfer from other tasks.(Reviewer `Uiyw`)
5. We have explored MSU relationships across tasks: In **Appendix J**, we investigate whether MSUs reflect semantic structures by projecting MSUs from 48 LoRAs (across different domains and tasks) into a shared space using t-SNE. The results show that MSUs from the same task and task cluster tend to be more closely aligned. (Reviewer `9LLA`)

These changes have been highlighted in blue font in the revision. We have also provided **point-by-point responses** to each reviewer's specific concerns. We believe these detailed revisions further demonstrate our contributions and address the reviewers' concerns.

Once again, we highly value the reviewers' insightful feedback and are open to any further discussions.

[1] Yu, Le, et al. "Language models are super mario: Absorbing abilities from homologous models as a free lunch." *Forty-first International Conference on Machine Learning*. 2024.

[2] Deep, Pala Tej, Rishabh Bhardwaj, and Soujanya Poria. "DELLA-Merging: Reducing Interference in Model Merging through Magnitude-Based Sampling." *arXiv preprint arXiv:2406.11617* (2024).

---

> ### Author Response · Authors · 2024-11-22
> **Any New Comments are Welcomed**
>
> Dear Reviewers,
>
> Thank you for your invaluable feedback on our paper. Your comments and suggestions have been very helpful in improving the clarity and quality of our work.
>
> As the discussion period deadline approaches, we would like to confirm that we have addressed all your concerns and resolved any remaining issues. We are also open to any additional comments or suggestions you might have and are happy to make further clarification if needed.
>
> Thank you once again for your time, thoughtful input, and dedication throughout this process.
>
> Best regards,
>
> The Authors

---

### Meta-Review · Area_Chair_gEXW · 2024-12-20

**Metareview:**

The paper introduces the concept of Minimal Semantic Unit (MSU) for disassembling and reassembling multiple LoRAs at a finer granularity, thereby ameliorating parameter interference and performance degradation. Evaluation results show the proposed MSU-based approach, combined with clustering to control the rank, yields to performance improvements over several benchmarks.

Reviewers generally found the approach interesting, but wanted to see comparisons to more SoTA competitors. The authors addressed this sufficiently during the rebuttal phase, introducing two very recent additional baselines (DARE and DELLA-Merging), to provide more comprehensive comparisons with recent model merging methods. A weakness that remained somewhat was the sensitivity to hyperparameters.

**Additional Comments On Reviewer Discussion:**

Beyond the two added baselines, reviewers appreciated additional experiments with more diverse tasks, the included assessment of scalability, the study of MSU similarity and the relationships across tasks. All these are worthy additions that have improved the paper.

---

### Decision · Program_Chairs · 2025-01-22

Accept (Poster)